# Molecular mechanism of choline and ethanolamine transport in humans

Keiken Ri[1,2,3,4,5,16], Tsai-Hsuan Weng[6,16], Ainara Claveras Cabezudo[7,8,16], Wiebke Jösting[6], Yu Zhang[1], Andre Bazzone[9], Nancy C. P. Leong[1,2,3,4,5], Sonja Welsch[10], Raymond T. Doty[11], Gonca Gursu[6], Tiffany Jia Ying Lim[1,2,3,4,5], Sarah Luise Schmidt[6], Janis L. Abkowitz[11], Gerhard Hummer[7,12 ✉], Di Wu[6,13 ✉], Long N. Nguyen[1,2,3,4,5 ✉] & Schara Safarian[6,13,14,15 ✉]

Human feline leukaemia virus subgroup C receptor-related proteins 1 and 2 (FLVCR1 and FLVCR2) are members of the major facilitator superfamily[1]. Their dysfunction is linked to several clinical disorders, including PCARP, HSAN and Fowler syndrome[2–7]. Earlier studies concluded that FLVCR1 may function as a haem exporter[8–12], whereas FLVCR2 was suggested to act as a haem importer[13], yet conclusive biochemical and detailed molecular evidence remained elusive for the function of both transporters[14–16]. Here, we show that FLVCR1 and FLVCR2 facilitate the transport of choline and ethanolamine across the plasma membrane, using a concentration-driven substrate translocation process. Through structural and computational analyses, we have identified distinct conformational states of FLVCRs and unravelled the coordination chemistry underlying their substrate interactions. Fully conserved tryptophan and tyrosine residues form the binding pocket of both transporters and confer selectivity for choline and ethanolamine through cation–π interactions. Our findings clarify the mechanisms of choline and ethanolamine transport by FLVCR1 and FLVCR2, enhance our comprehension of disease-associated mutations that interfere with these vital processes and shed light on the conformational dynamics of these major facilitator superfamily proteins during the transport cycle.

The feline leukaemia virus subgroup C receptor (FLVCR) family, a member of the major facilitator superfamily (MFS) of secondary active transporters, consists of four paralogues encoded by the human *SLC49* gene group[1]. FLVCR1 (also known as SLC49A1 or MFSD7B) was initially identified as the cell surface receptor for feline leukaemia virus (FeLV)[17]. FLVCR2 (also known as SLC49A2 or MFSD7C) shares 60% sequence identity with FLVCR1 in the transmembrane domain but does not bind the feline leukaemia virus subgroup C envelope protein[18]. Both transporters exhibit ubiquitous tissue distribution in humans and have substantial haematopathological and neuropathological implications[1,16]. Dysfunction of FLVCR1 caused by germline mutations is associated with posterior column ataxia with retinitis pigmentosa (PCARP)[2,3] and hereditary sensory and autonomic neuropathies (HSAN)[4,5]. Similarly, truncation and missense mutations in *FLVCR2* are associated with autosomal-recessive cerebral proliferative vasculopathy (Fowler syndrome)[6,7]. Furthermore, both FLVCR variants are suggested to have a key role in cell development and differentiation, including angiogenesis and tumorigenesis[19–22].

Earlier studies demonstrated that FLVCR1 was necessary and sufficient to regulate cellular haem content and thus concluded that FLVCR1 may function as a putative haem exporter[8–12], whereas FLVCR2 was suggested to act as a haem importer[13], yet their definitive roles in this capacity remain elusive[14–16]. To understand their functions, experimental validation at the biochemical and molecular levels is necessary, which will connect the physiological roles of these transporters and clinical relevance to their specific mechanistic actions. Recent studies indicated that FLVCR1 is involved in choline transport; however, the ligands for FLVCR2 remain elusive[23]. Here, we used an integrative approach, including cell-based radioligand transport assays, single-particle analysis cryo-electron microscopy (cryo-EM), structure-guided mutagenesis and atomistic molecular dynamics simulations to characterize the ligand specificity, molecular architecture and the conformational landscape of FLVCR1 and FLVCR2 transporters.

## Substrate preference of FLVCR1 and FLVCR2

We overexpressed the human *FLVCR1* and *FLVCR2* genes in human embryonic kidney (HEK293) cells to substantiate and characterize their roles in cellular choline and ethanolamine transport[24,25].

[1]Department of Biochemistry, Yong Loo Lin School of Medicine, National University of Singapore, Singapore, Singapore. [2]Immunology Program, Life Sciences Institute, National University of Singapore, Singapore, Singapore. [3]Singapore Lipidomics Incubator (SLING), Life Sciences Institute, National University of Singapore, Singapore, Singapore. [4]Cardiovascular Disease Research (CVD) Programme, Yong Loo Lin School of Medicine, National University of Singapore, Singapore, Singapore. [5]Immunology Translational Research Program, Yong Loo Lin School of Medicine, National University of Singapore, Singapore, Singapore. [6]Department and Emeritus Group of Molecular Membrane Biology, Max Planck Institute of Biophysics, Frankfurt, Germany. [7]Department of Theoretical Biophysics, Max Planck Institute of Biophysics, Frankfurt, Germany. [8]IMPRS on Cellular Biophysics, Frankfurt, Germany. [9]Nanion Technologies GmbH, Munich, Germany. [10]Central Electron Microscopy Facility, Max Planck Institute of Biophysics, Frankfurt, Germany. [11]Division of Hematology, Department of Medicine, University of Washington, Seattle, WA, USA. [12]Institute of Biophysics, Goethe University Frankfurt, Frankfurt, Germany. [13]Fraunhofer Institute for Translational Medicine and Pharmacology ITMP, Frankfurt, Germany. [14]Institute of Clinical Pharmacology, Faculty of Medicine, Goethe University Frankfurt, Frankfurt, Germany. [15]Fraunhofer Cluster of Excellence for Immune Mediated Diseases (CIMD), Frankfurt, Germany. [16]These authors contributed equally: Keiken Ri, Tsai-Hsuan Weng, Ainara Claveras Cabezudo. ✉e-mail: gerhard.hummer@biophys.mpg.de; di.wu@biophys.mpg.de; bchnnl@nus.edu.sg; schara.safarian@biophys.mpg.de

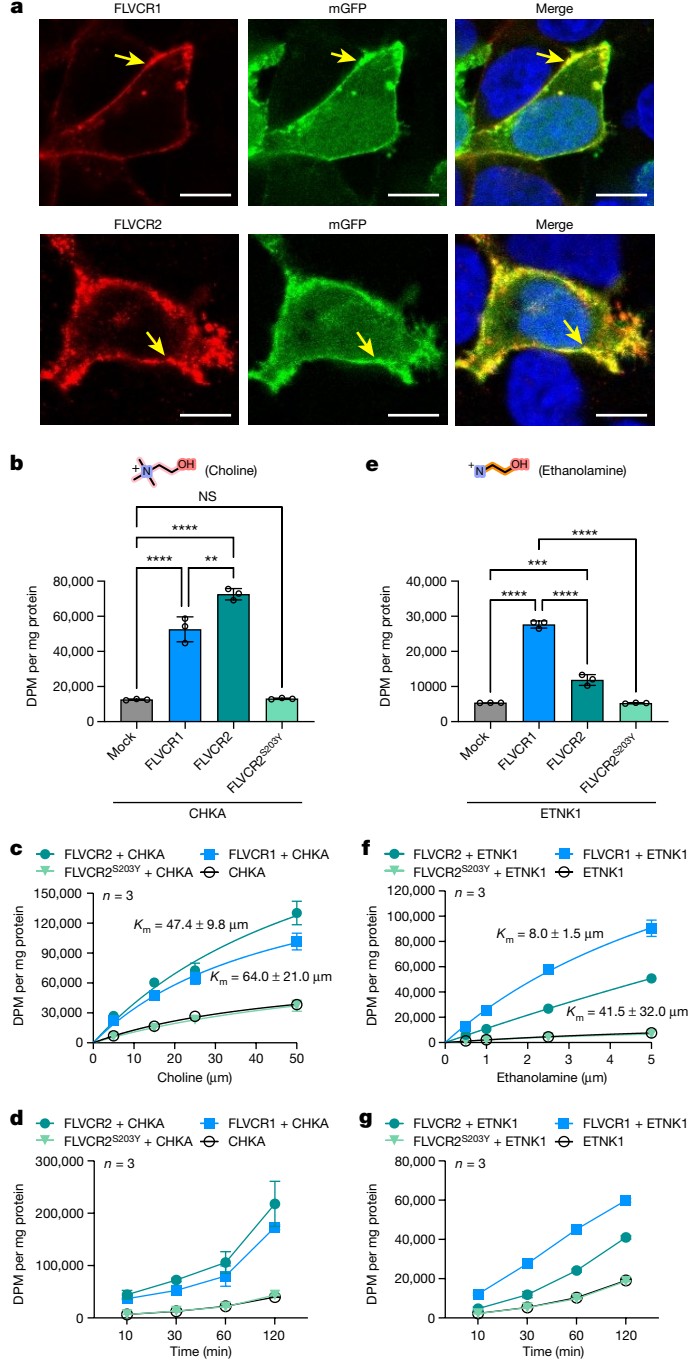

**Fig. 1 | FLVCR1 and FLVCR2 are choline and ethanolamine transporters.**
**a**, Confocal imaging shows that FLVCR1 and FLVCR2 are localized at the plasma membrane (arrows). Plasma membrane GFP (mGFP) was used as a marker. Each experiment was performed at least three times independently. Representative images are shown. **b**, Choline transport activities of human FLVCR1 and FLVCR2. CHKA was co-expressed with both proteins. DPM, disintegrations per minute. **c,d**, Dose curves (**c**) and time courses (**d**) for choline transport activities of human FLVCR1 and FLVCR2. CHKA was co-expressed with both proteins. Experiments were repeated twice on different days. A second dataset is also provided in the Source Data. **e**, Ethanolamine transport activities of human FLVCR1 and FLVCR2. ETNK1 was co-expressed with both proteins. **f,g**, Dose curves (**f**) and time courses (**g**) for ethanolamine transport activities of human FLVCR1 and FLVCR2. ETNK1 was co-expressed with both proteins. Experiments were repeated twice on different days. A second dataset is also provided in the Source Data. For **b**–**g**, $n = 3$ biologically independent samples (wells). The inactive S203Y mutant of FLVCR2 and empty vector (mock) were used as controls. Data are expressed as mean ± s.d. In **b**, ****$P < 0.0001$, **$P = 0.0011$. In **e**, ****$P < 0.0001$, ***$P = 0.0001$. NS, not significant. One-way analysis of variance (ANOVA) for transport activity measurement; two-way ANOVA for dose curve measurements. Note that the dataset used in **b** and **e** was the same dataset from **d** and **g** at 30 min time point, respectively.

in *FLVCR1*-knockout livers (Extended Data Fig. 2b,c). Subsequent cell-based assays revealed that both FLVCRs facilitate ethanolamine uptake into cells (Extended Data Fig. 1b). Notably, co-expression of ethanolamine kinase 1 (*ETNK1*) enhanced the ethanolamine transport rate of FLVCR1 fivefold but did not substantially affect the efficiency of FLVCR2 (Fig. 1e–g and Extended Data Fig. 1b). We determined transport kinetic parameters of FLVCR1 and FLVCR2, giving $K_m$ values of 47.4 ± 9.8 and 64.0 ± 21.0 µM for choline and 8 ± 1.5 and 41.5 ± 32.0 µM for ethanolamine under this tested condition, respectively (Fig. 1c,f). To decipher the transport mechanism of both FLVCRs, we investigated their reliance on sodium ions and pH levels. Our uptake studies revealed that FLVCR-mediated transport of choline and ethanolamine is not contingent on sodium ion involvement and operates effectively across a broad pH range (Extended Data Fig. 3a,b). This finding underscores that neither sodium nor pH gradients are critical for the translocation of choline and ethanolamine by FLVCRs. We further assessed the mechanistic properties of FLVCR2 by performing a choline washout assay. On inverting the choline gradient across the plasma membrane, we measured a significant decrease of cellular choline levels within 1 h, indicating a bidirectional choline transport activity mediated by FLVCR2 (Extended Data Fig. 3c). FLVCR1 showed similar properties in an ethanolamine washout experiment (Extended Data Fig. 3c). Our findings suggest that both FLVCR1 and FLVCR2 operate as uniporters, facilitating downhill ligand transport independent of sodium or pH gradients.

## Overall architecture of FLVCR1 and FLVCR2

Next, we aimed to establish structure–function relationships for the choline and ethanolamine transport properties of FLVCR1 and FLVCR2. Wild-type FLVCR1 and FLVCR2 proteins were purified and subjected to single-particle analysis cryo-EM (Supplementary Fig. 2e,f). We determined structures of FLVCR1 and FLVCR2 in as-isolated (without exogenous addition of ligands), inward-facing conformations at 2.9 Å resolution (Fig. 2a,b and Supplementary Figs. 3 and 4). In addition, we elucidated the structure of FLVCR2 in an outward-facing conformation at 3.1 Å resolution from the same sample preparation (Fig. 2c and Supplementary Fig. 3). Both FLVCR paralogues share a common MFS-type architecture[26], with their N domain containing transmembrane helices 1–6 (TM1–6) and the C domain containing TM7–12 connected by a long and flexible loop containing two short horizontal cytoplasmic helices (H1 and H2) (Extended Data Fig. 4a,b). In the FLVCR2 structure, we resolved a short helical segment at the C terminus (H3), whereas the

On overproduction, FLVCR1a (hereafter referred to as FLVCR1) and FLVCR2 localized at the plasma membrane of HEK293 cells (Fig. 1a and Supplementary Fig. 2a–d). Radioactive [³H]choline transport assays showed a discernible increase in uptake facilitated by FLVCR2, whereas FLVCR1 did not exhibit such an effect under the tested condition (Extended Data Fig. 1a). Notably, co-expression of the choline kinase A (*CHKA*) gene significantly enhanced choline uptake by both transporters in dose- and time-dependent manners (Fig. 1b–d). This suggests that choline influx mediated by FLVCR1 and FLVCR2 is enhanced by the overproduction of downstream choline-using enzymes. To ascertain whether choline is the principal physiological substrate for FLVCR1, we performed a comprehensive metabolomic analysis on liver samples from *FLVCR1*-knockout mice (Extended Data Fig. 2a). This led us to discover that, in addition to choline and its metabolites, ethanolamine metabolite profiles were also affected

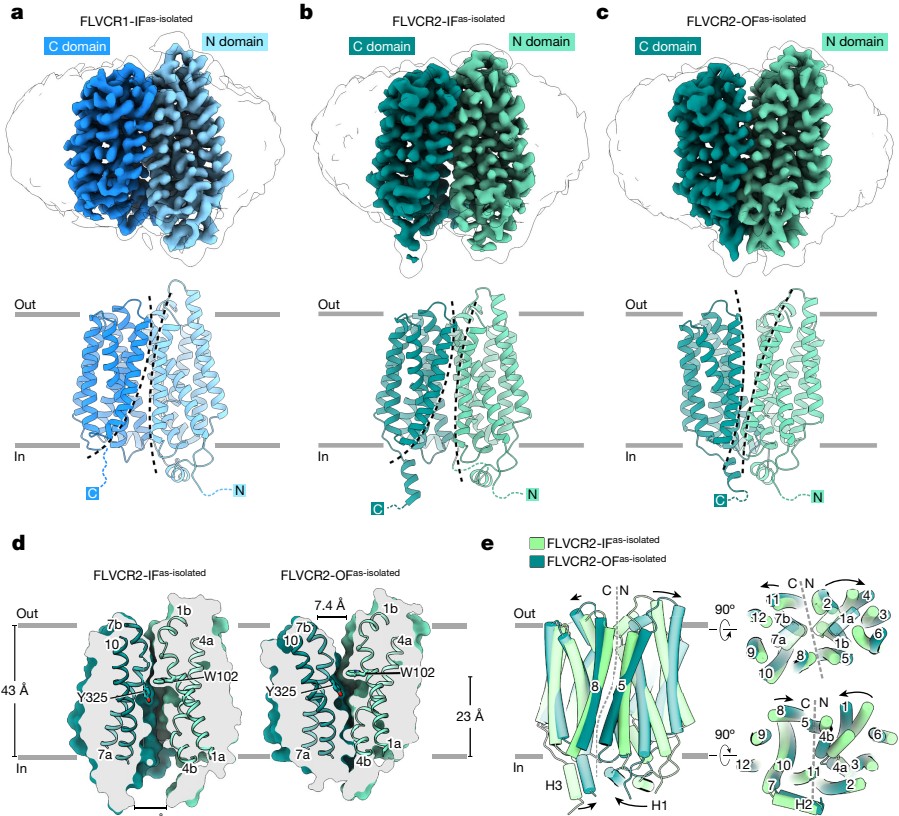

**Fig. 2 | Architecture of FLVCR1 and FLVCR2 in their inward- and outward-facing states. a–c**, Cryo-EM density (top) and atomic model (middle and bottom) of as-isolated, inward-facing FLVCR1 (FLVCR1-IF^as-isolated) (**a**) as well as FLVCR2 in the inward-facing (FLVCR2-IF^as-isolated) (**b**) and outward-facing (FLVCR2-OF^as-isolated) (**c**) states. The N and C domains are coloured in different shades of blue and green for FLVCR1 and FLVCR2, respectively. A transparent cryo-EM density lowpass-filtered at 6 Å is shown to visualize the detergent belt surrounding the transmembrane region. **d**, Cut-away views of the surface representation showing the cavity shape of FLVCR2-IF^as-isolated (left) and FLVCR2-OF^as-isolated (right). Two central aromatic residues are shown as sticks. **e**, Structural superposition of FLVCR2-OF^as-isolated (dark green) and FLVCR2-IF^as-isolated (light green).

density of the FLVCR1 C terminus was less pronounced (Extended Data Fig. 4a,b and Supplementary Fig. 5a–c).

The inward-facing conformations of FLVCR1 and FLVCR2 exhibit a close resemblance to each other, with a root mean square deviation (Cα r.m.s.d.) of 0.993 Å (Extended Data Fig. 4b). Both structures feature a wedge-shaped solvent-accessible cavity that is mainly created by the separation of TM4 and TM5 of the N domain from TM10 and TM11 of the C domain (Fig. 2d and Extended Data Fig. 4b,c). This space extends halfway across the membrane with a similar depth of around 23 Å in both transporters. On the level of the outer leaflet, TM1, TM2 and TM5 of the N domain and TM7, TM8 and TM11 of the C domain pack tightly against each other and thus shield the central cavity from the extracellular space (Fig. 2d and Extended Data Fig. 4b,c). Notably, TM1, TM4 and TM7 exhibit disordered regions at their C-terminal ends, followed by short kinked helical motifs designated TM1b, TM4b and TM7b (Fig. 2d and Extended Data Fig. 4b,c).

The outward-facing conformation of FLVCR2 features a cavity accessible from the extracellular space. This cavity is formed through a 'rocker-switch' rigid-body motion that occurs during the transition from the inward-facing to the outward-facing state (Fig. 2d,e and Supplementary Video 1). This motion involves a shift of the outer halves of all transmembrane segments away from the central axis. Concurrently, the inner halves of these TMs move inward, effectively blocking the exit pathway (Fig. 2e). The cavities in outward- and inward-facing conformations are 7.4 Å and 8.6 Å wide at their respective openings (Fig. 2d). Both inward- and outward-facing cavities are lined by uncharged and

hydrophobic residues that are mostly conserved (Extended Data Fig. 5). Compared to the inward-facing cavity, the outward-facing cavity demonstrates a more restricted pathway within its central region. In this narrowed segment of the cavity, the channel ends at W102^FLVCR2 and Y325^FLVCR2 (equivalent to W125^FLVCR1 and Y349^FLVCR1), two residues that are highly conserved in both FLVCR transporters (Fig. 2d and Extended Data Figs. 4c and 5).

In line with an alternating-access model, the inward-facing conformation of FLVCR2 features a tightly sealed extracellular gate created by interdomain interactions. This is mainly achieved by the juxtaposition of TM1b (N domain) and TM7b (C domain) (Fig. 2d and Extended Data Fig. 4b). The interdomain interaction between these two motifs is stabilized by a hydrogen-bonding network consisting of two pseudosymmetry-related asparagine residues N110^FLVCR2 and N332^FLVCR2 and E343^FLVCR2 (TM8) and N239^FLVCR2 located on the extracellular loop connecting TM5 and TM6 (EL5–6) (Extended Data Fig. 4c and Supplementary Fig. 6a). Furthermore, we identified a stable interdomain salt bridge between D124^FLVCR2 (TM2) and R333^FLVCR2 (TM7b), reinforcing the external occlusion (Extended Data Fig. 4c and Supplementary Fig. 6b). Our cell-based mutagenesis studies show that alanine substitutions at N110^FLVCR2, E343^FLVCR2, D124^FLVCR2 and R333^FLVCR2 individually result in a significant decrease of choline uptake and an almost complete perturbation of ethanolamine transport, highlighting a critical role of interdomain interactions on the extracellular surface for FLVCR2 functionality (Extended Data Fig. 4d and Supplementary Fig. 7). Although FLVCR1 exhibits a similar extracellular

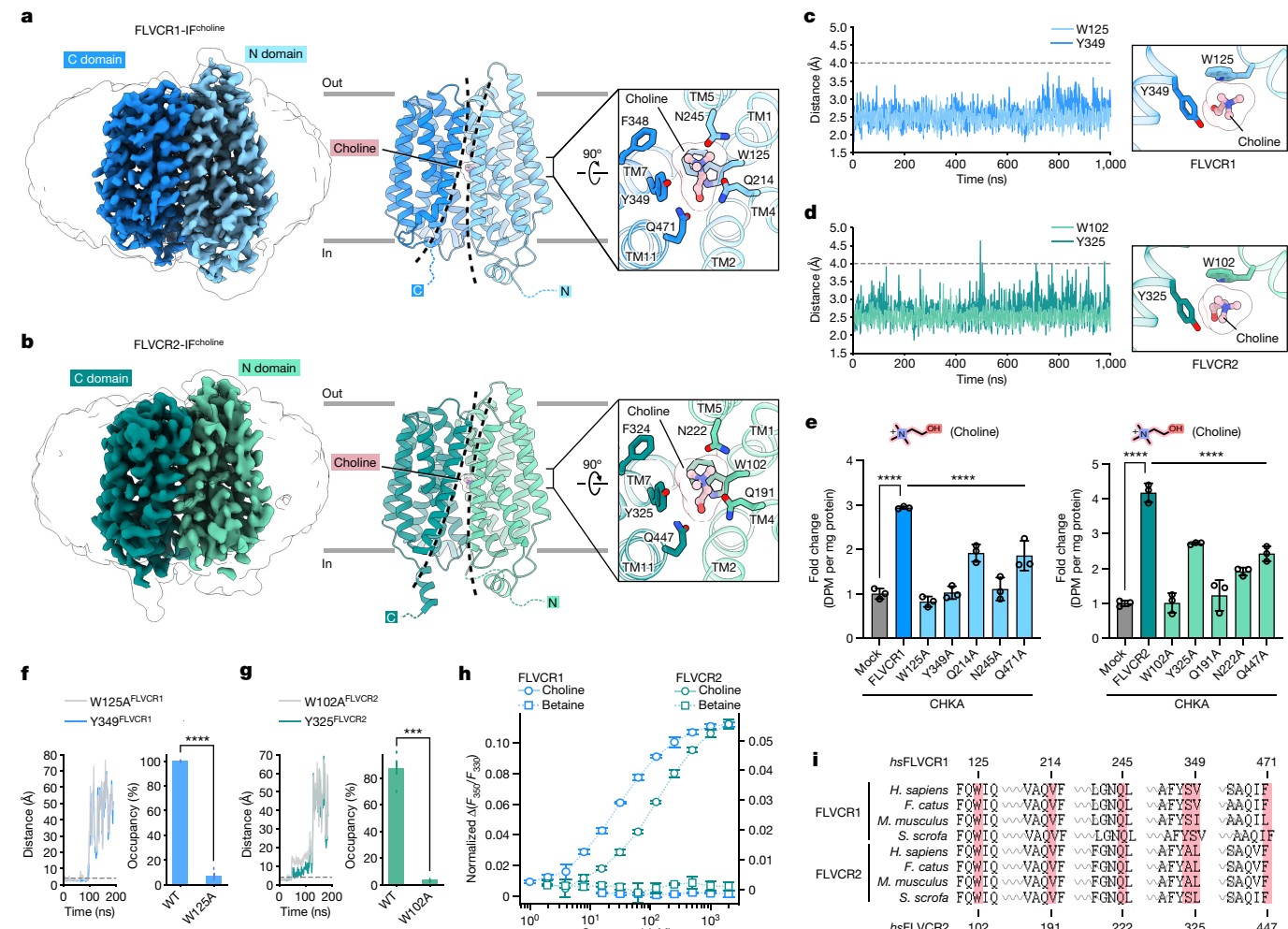

**Fig. 3 | Cryo-EM structures of FLVCR1 and FLVCR2 in complex with choline.**
**a,b**, Cryo-EM densities and atomic models of the choline-bound inward-facing FLVCR1 (FLVCR1-IF$^{choline}$) structure (**a**) and the choline-bound inward-facing FLVCR2 (FLVCR2-IF$^{choline}$) structure (**b**), respectively. The bound choline is shown as ball-and-stick model; binding-site residues are shown as sticks.
**c,d**, Minimum atom-pair distances between choline and the highly conserved tryptophan and tyrosine side chains forming the choline-binding pockets of wild-type FLVCR1 (**c**) and FLVCR2 (**d**) in 1 μs molecular dynamics simulations. Distances less than 4 Å indicate cation–π interactions (grey dashed line).
**e**, Choline transport activity of indicated FLVCR1 and FLVCR2 mutants. CHKA was co-expressed with all proteins. Respectively, 20 μM [³H]choline was used for FLVCR1 mutants and 100 μM [³H]choline was used for FLVCR2 mutants. Mutant transport activities were normalized to the total protein from cells and shown as fold change with reference to mock (empty vector). $n$ = 3 biologically independent samples (wells). **f,g**, Escape of choline from the binding site of W125A$^{FLVCR1}$ (**f**) and W102A$^{FLVCR2}$ (**g**) mutant variants. Shown are distances as function of time in molecular dynamics simulations (left) and choline occupancy in binding site for wild-type (WT) and alanine mutants (right). **h**, Shifts of tryptophan fluorescence of FLVCR1 and FLVCR2 in the presence of choline or betaine. $n$ = 3 independent technical replicates. **i**, Protein sequence alignment of choline-binding-pocket residues (red block) in FLVCR1 and FLVCR2 across various mammalian species. Indicated residue numbers refer to FLVCR1 and FLVCR2 from *Homo sapiens*. Data shown are mean ± s.d. for **e** and **h** and mean ± s.e.m. for **f** and **g**. In **e**, ****$P$ < 0.0001. In **f** and **g**, ****$P$ < 0.0001, ***$P$ = 0.00068. One-way ANOVA for **e** and unpaired two-tailed $t$-test for **f** and **g**.

hydrogen-bonding network as FLVCR2, it lacks a corresponding salt bridge interaction (Extended Data Fig. 4d). Notably, the disruption of the hydrogen bond between N133$^{FLVCR1}$ and N359$^{FLVCR1}$ (structurally analogous to N110$^{FLVCR2}$ and N332$^{FLVCR2}$) through an alanine substitution of N133$^{FLVCR1}$ markedly diminishes the transport activity of FLVCR1 for both choline and ethanolamine. By contrast, the E367A$^{FLVCR1}$ mutation impacts the transport of choline to a stronger extent compared with ethanolamine transport (Extended Data Fig. 4e).

During the transition from inward- to outward-facing conformation of FLVCR2, interdomain interactions contributing to the extracellular gate become disrupted, whereas the formation of the intracellular gate occludes the central cavity from the cytoplasmic side. TM4b of the N domain moves into proximity to the N-terminal end of TM11 of the C domain to establish a first level of occlusion (Fig. 2e). An interaction network consisting of several hydrogen bonds and a salt bridge is

found within this region as well (Extended Data Fig. 4c). The E435$^{FLVCR2}$ residue in TM11 has a pivotal and multifaceted role, forming a hydrogen bond with S203$^{FLVCR2}$ and simultaneously establishing a salt bridge with R200$^{FLVCR2}$ in TM4b (Extended Data Fig. 4c and Supplementary Fig. 6a,b). Individuals with a homozygous S203Y mutation in FLVCR2 are non-viable, which is probably attributable to the complete loss of choline and ethanolamine transport activity[27] (Fig. 1b–g). Analogously, a S203A mutant also exhibited abolished transport activity (Extended Data Fig. 4d). Our mutagenesis studies underscore a greater significance of S203$^{FLVCR2}$ compared to R200$^{FLVCR2}$, particularly in choline transport (Fig. 1b and Extended Data Fig. 4d). This suggests that the interaction between E435$^{FLVCR2}$ and S203$^{FLVCR2}$ is essential for either facilitating conformational changes during the transport cycle or maintaining the stability of FLVCR2 in its outward-facing state (Supplementary Fig. 6c). Another interdomain hydrogen bond is identified

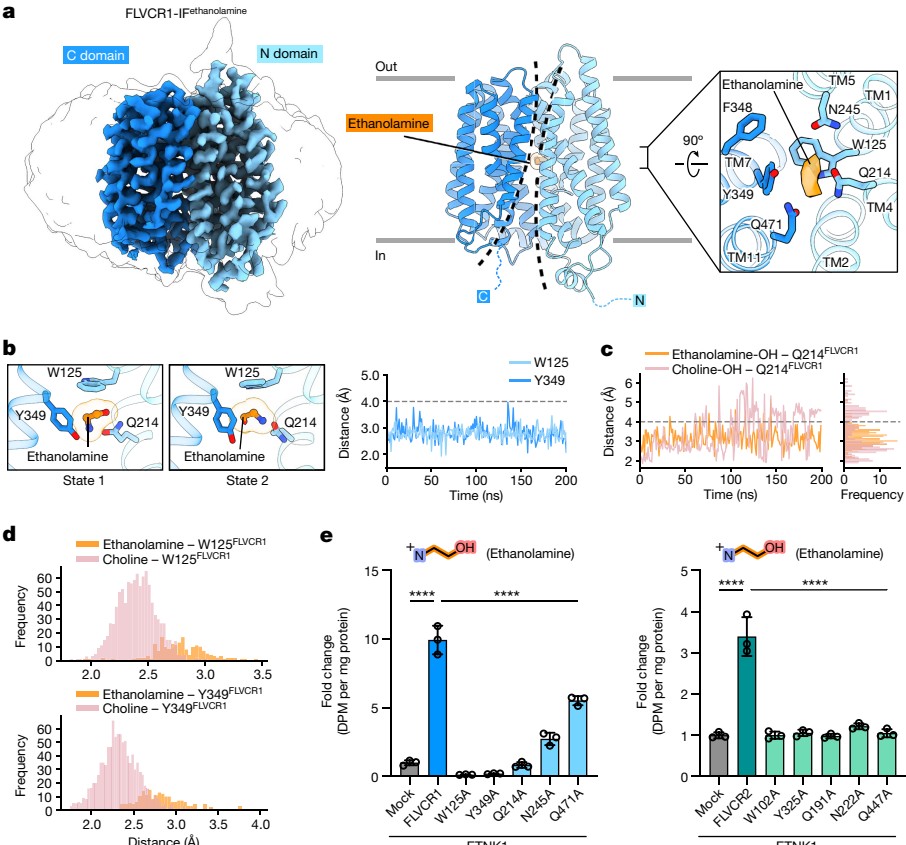

**Fig. 4 | Cryo-EM structures of FLVCR1 in complex with ethanolamine.**
**a**, Cryo-EM density and atomic model of the ethanolamine-bound inward-facing FLVCR1 (FLVCR1-IF^ethanolamine) structure. The identified ethanolamine density within the ligand-binding pocket is shown in orange; binding-site residues are shown as sticks. **b**, Snapshots of the two distinct states of ethanolamine (left) and minimum atom-pair distance of ethanolamine to conserved aromatic side chains of FLVCR1 as function of time in molecular dynamics simulations (right). **c**, Minimum atom-pair distance between Q214^FLVCR1 and the hydroxyl group of ethanolamine (orange) or choline (pink) as function of time in molecular

dynamics simulations, with corresponding frequency (right). **d**, Frequency of minimum atom-pair distance between ethanolamine or choline and W125^FLVCR1 (top) or Y349^FLVCR1 (bottom) derived from Figs. 3c and 4b, respectively. **e**, Ethanolamine transport activities of indicated FLVCR1 and FLVCR2 mutants. ETNK1 was co-expressed with all proteins. We used 2.5 μM [$^{14}$C]ethanolamine. Transport activities of mutant variants were normalized to the total protein from cells and are shown as fold change to mock (empty vector). $n = 3$ biologically independent samples (wells). Data shown are mean ± s.d. In **e**, ****$P < 0.0001$. One-way ANOVA for **e**.

between S199^FLVCR2 in TM4b and S439^FLVCR2 in TM1 (Extended Data Fig. 4c and Supplementary Fig. 6a,c). In the peripheral region, K372^FLVCR2 and R374^FLVCR2 (EL8–9) approach N209^FLVCR2 (EL4–5) and S212^FLVCR2 (TM5) to form hydrogen bond pairs and thus block the lateral accessibility of the cavity (Extended Data Fig. 4c and Supplementary Fig. 6a,c). A second level of occlusion was observed beneath the cytoplasmic ends of TM4, TM10 and TM11, where H1 and H3 are positioned in close proximity (Extended Data Fig. 4c). Here, the backbone carbonyl group and amide nitrogen of A283^FLVCR2 (H1) form stable hydrogen bonds with N497^FLVCR2 (H3) and Y431^FLVCR2 located on the intracellular loop connecting TM10 and TM11 (IL10–11), respectively (Extended Data Fig. 4c and Supplementary Fig. 6a,c). Together with the loop connecting TM10 and TM11, the two helical motifs H1 and H3 serve as a cytoplasmic latch to secure the closure of the two domains.

## Choline coordination by FLVCR1 and FLVCR2

To decipher the substrate binding and coordination chemistry of FLVCR1 and FLVCR2, we determined their structures with choline bound at resolutions of 2.6 Å and 2.8 Å, respectively (Fig. 3a,b and Supplementary Figs. 3, 4 and 5d,e). Both structures were captured in the inward-facing conformation, suggesting that choline is capable of resolving the previously captured conformational heterogeneity

of FLVCR2. The choline-bound structures exhibit a Cα r.m.s.d. of 0.688 Å (FLVCR1) and 1.002 Å (FLVCR2), respectively, from their as-isolated, inward-facing structures (Extended Data Fig. 6a,b). In the choline-bound structures, the binding sites are located at analogous positions, with the choline molecule situated between the two domains, surrounded mainly by TM1, TM2, TM4, TM5, TM7 and TM11 (Fig. 3a,b). The residues forming the binding site in the two transporters are conserved. We observed that W125^FLVCR1 and W102^FLVCR2 in TM1 directly interact with choline. This central coordinating tryptophan residue is located above choline, protruding its side chain to constrain the diffusion of the molecule towards the extracellular space (Figs. 2d and 3a,b). Two more aromatic residues of TM7, one tyrosine (Y349^FLVCR1 and Y325^FLVCR2) and one phenylalanine (F348^FLVCR1 and F324^FLVCR2), line the peripheral space of the binding site and restrict the movement of choline within the pocket (Fig. 3a,b). Our molecular dynamics simulations demonstrate that the quaternary ammonium group of choline forms stable cation–π interactions with the conserved tryptophan (W125^FLVCR1 and W102^FLVCR2) and tyrosine residues (Y349^FLVCR1 and Y325^FLVCR2) in a simultaneous manner (Fig. 3c,d). Mutations of W125A^FLVCR1 and W102A^FLVCR2 significantly reduced choline transport activity (Fig. 3e and Supplementary Fig. 7). Molecular dynamics simulations further confirmed the key role of the tryptophan in both transporters (Fig. 3f,g). Free energy calculations using molecular

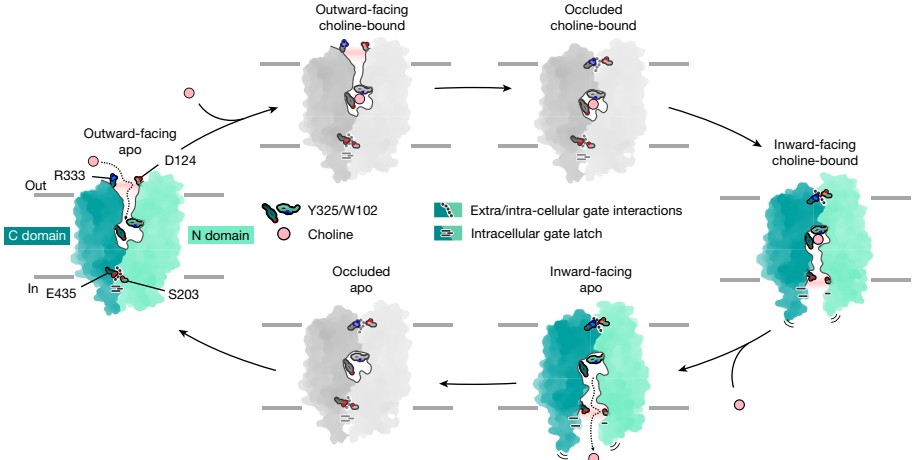

**Fig. 5 | Proposed model for choline transport by FLVCR2.** Schematic illustration of FLVCR2 conformations during the choline transport cycle. Green-coloured states represent experimentally obtained conformations in this study. States coloured in grey are hypothesized on the basis of knowledge about the commonly characterized alternative-access mechanism of MFS transporters.

mechanics/Poisson–Boltzmann surface area (MM/PBSA) support these observations (Extended Data Fig. 7a,b). We also found that the hydroxyl group of choline is essential for ligand recognition by both FLVCRs. Substitution to a carboxylic group, as found in betaine, abolishes ligand binding and transport activity[25] (Fig. 3h). In our choline simulations, the hydroxyl group shows versatile interactions by forming transient hydrogen bonds with two proximal glutamine residues (Q214[FLVCR1]/Q471[FLVCR1] and Q191[FLVCR2]/Q447[FLVCR2]). Further interactions come from at least two water molecules, which were consistently present around the hydroxyl moiety of the choline (Supplementary Fig. 8 and Supplementary Video 2). Functional analyses of Q214[FLVCR1], Q471[FLVCR1], Q191[FLVCR2] and Q447[FLVCR2] indicate that changes of the local protein environment around the hydroxyl group of choline, caused by alanine substitutions, discernibly affect the transport activities of both FLVCRs (Fig. 3e). It is noteworthy that all these residues are fully conserved in mammalian FLVCR homologues, suggesting a common substrate profile of FLVCRs across species (Fig. 3i).

## Ethanolamine coordination by FLVCR1

Guided by our functional analyses, we advanced our investigation into the molecular mechanism of ethanolamine binding by FLVCR1. We obtained the cryo-EM structure of FLVCR1 complexed with ethanolamine at a resolution of 2.9 Å, showing an inward-facing conformation analogous to the choline-bound FLVCR1 structure (Fig. 4a, Extended Data Fig. 6c,d and Supplementary Fig. 3). The ethanolamine density was identified within the above-mentioned ligand-binding pocket, occupying the same cleft as choline (Fig. 4a and Supplementary Fig. 5d–f). To characterize the coordination and binding dynamics, we carried out molecular dynamics simulations with ethanolamine in the binding site. In our simulations, ethanolamine exhibited pronounced dynamics, adopting different poses with two distinct binding state orientations (Fig. 4b and Extended Data Fig. 8a). In both states, the hydroxyl group of ethanolamine interacted with Q214[FLVCR1] and the primary amine group of ethanolamine largely maintained cation–π contacts with the conserved aromatics W125[FLVCR1] and Y349[FLVCR1] (Fig. 4c,d and Extended Data Fig. 8b,c). Such interactions are also reflected in our MM/PBSA calculations (Extended Data Fig. 7c). Our molecular dynamics analyses indicate that the binding of choline in FLVCR1 is characterized by greater stability, probably due to the steady interaction of its quaternary ammonium group with surrounding coordination residues. These nuanced differences in their molecular mechanism presumably contribute to the selective substrate

profile of FLVCR1 as observed in our cell-based transport assays in which FLVCR1 showed a preferential ethanolamine transport activity (Figs. 1b,e, 3e and 4e). This is further substantiated by the Q214A[FLVCR1] mutant, which results in a complete loss of ethanolamine transport activity while only partially affecting choline transport (Figs. 3e and 4e). Notably, alanine substitution of the analogous Q191[FLVCR2] residue abolishes the transport of both ethanolamine and choline in FLVCR2 (Figs. 3e and 4e).

Of note, although the as-isolated, inward-facing and outward-facing conformations of FLVCR2 represent apo states of the transporter, we observed an extra non-protein density within the substrate-binding cleft of the as-isolated FLVCR1 structure (Supplementary Figs. 5a–c and 9). Introducing choline and ethanolamine to FLVCR1 leads to displacement of this unidentified molecule, evident from the altered size and shape of the specific densities for choline and ethanolamine in the respective electron microscopy maps (Supplementary Fig. 9). The precise nature of this ligand remains unknown. It is, however, conceivable that FLVCR1 is copurified with a blend of choline and ethanolamine, creating an indistinct density for these ligands or alternatively with a yet-unidentified ligand.

## Translocation pathways in FLVCRs

MFS transporters typically cycle between inward- and outward-facing conformations, facilitating substrate translocation in an alternating-access manner and substrate binding has a pivotal role in eliciting the conformational transitions[28]. Our cryo-EM data support this mechanism in FLVCR2, in which we see a full transition from the outward-facing to the inward-facing state on choline-binding. Complementary to our structural insights into the conformational landscape, we performed molecular dynamics simulations to map the route for choline entry along the pathway in the outward-facing conformation of FLVCR2. After spontaneously diffusing into the translocation pathway, choline initially interacts with several residues near the protein surface, mainly D124[FLVCR2] (Supplementary Fig. 10). It sequentially approaches the deeper recesses of the cavity, primarily engaging with conserved aromatic residues W102[FLVCR2] and Y325[FLVCR2]. In one of the entry events, we observed choline moving spontaneously to a position below the W102[FLVCR2] residue within the binding site consistent with our structural data (Supplementary Video 3).

The substantial global conformational changes triggered by substrate further alter the local arrangement of substrate-coordinating residues within the translocation pathway as seen in our cryo-EM maps.

The rearrangement of the binding site results in a more constricted pocket, promoted by the inward movement of the conserved residues towards the substrate molecule (Extended Data Fig. 9a). The repositioning of these residues, especially the conserved aromatic side chains of W102$^{FLVCR2}$, F324$^{FLVCR2}$ and Y325$^{FLVCR2}$ (equivalent to W125$^{FLVCR1}$, F348$^{FLVCR1}$ and Y349$^{FLVCR1}$) restricts the accessibility of the binding pocket and promotes substrate engulfment and coordination. In their choline-bound structures, the cavities of both FLVCRs share common characteristics, exhibiting a neutral interior but a negatively charged surface at the exit (Extended Data Fig. 5b,e). FLVCR1 features a slightly smaller cavity volume of 513 Å$^3$ compared to 579 Å$^3$ of FLVCR2 (Extended Data Fig. 9b). Although the cavity narrows towards the intracellular side, a peripheral solvent-accessible channel emerging from the cytoplasmic space to the binding site reveals a semi-open translocation pathway in the substrate-bound, inward-facing conformations of both transporters, which may facilitate the release of the bound ligand (Extended Data Fig. 9b), as seen in molecular dynamics simulations (Fig. 3g and Supplementary Fig. 11b,c).

## Discussion

Although many studies point to a role for FLVCR1 in the regulation of cellular haem export[8–12], the study of FLVCR2-mediated haem uptake has not been confirmed[13] and, furthermore, conclusive biochemical and detailed molecular evidence has remained elusive for the function of both transporters[14–16]. In this study, we have investigated the choline and ethanolamine transport properties of FLVCR1 and FLVCR2 using cell-based transport assays and determined cryo-EM structures of these human FLVCRs in distinct as-isolated, choline-bound and ethanolamine-bound states (Extended Data Table 1). Our work provides insights into the architecture, ligand binding chemistry and conformational landscapes of these two MFS transporters. By means of several lines of evidence, we conclude that choline and ethanolamine represent primary transport substrates of both FLVCR1 and FLVCR2. Our findings indicate that both FLVCR1 and FLVCR2 operate as uniporters, facilitating downhill ligand transport independent of sodium or pH gradients. Notably, FLVCR1 and FLVCR2 possess similar ligand binding and coordination chemistries in which a key conserved tryptophan residue (W125$^{FLVCR1}$/W102$^{FLVCR2}$) forms a cation–π interaction with the ammonium and amine groups of the respective ligands. Further, we characterized the functional importance of several glutamine residues within the substrate-binding site that participate in hydrogen bonds with the hydroxyl groups of choline and ethanolamine during ligand coordination and transport. Functional assays point to a ligand preference between FLVCR1 and FLVCR2. Our structural studies suggest that ligand coordination is a crucial determinant of the transport preference between choline and ethanolamine for both FLVCRs. Given that most solute carrier transporters exhibit ligand promiscuity[29,30], the specific role of FLVCR1 as an ethanolamine transporter and FLVCR2 as a choline transporter at physiological conditions has yet to be confirmed through in vivo studies in animal models. Our recent study shows that FLVCR2 functions as a choline exporter at the blood–brain barrier[25].

On the basis of our structural findings and simulation data, we suggest a rocker-switch, alternating-access mechanism for the transport cycle of choline import (Fig. 5 and Supplementary Videos 3 and 4). We propose a transport cycle model for both FLVCRs in which the outward-facing conformation represents the state for ligand binding from the extracellular space. Substrate-induced conformational changes drive the transporter towards its inward-facing state. Finally, ligands are released to the intracellular space to enter metabolic pathways. Subsequent to choline release, FLVCRs will undergo further conformation changes to adopt their outward-facing state and re-initiate the transport cycle. Our structural data did not reveal fully occluded conformations that are key features of alternating access. Hence, we suspect that these occluded conformations exist only transiently and rapidly convert towards either the inward- or outward-facing conformations[28,31,32].

In summary, our research delineates the structural framework through which FLVCR1 and FLVCR2 mediate the cellular transport of choline and ethanolamine. Identifying these proteins as facilitative transporters provides a crucial groundwork for future characterizations of the physiological functions of choline and ethanolamine transport in vivo. Furthermore, our findings provide mechanistic insights for understanding disease aetiology pertaining to clinical mutations of the *SLC49A1* and *SLC49A2* genes encoding these two essential transporters.

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

# Methods

## Generation of inducible HEK293 stable cell lines

The complementary DNAs of full-length wild-type FLVCR1 (human SLC49A1, NCBI reference sequence NM_014053) and FLVCR2 (human SLC49A2, NCBI reference sequence NM_017791) were cloned into pcDNA5/FRT/TO (Invitrogen) vectors, respectively. The gene for both FLVCRs was modified by a C-terminal FLAG fusion tag. Further details are found in sequence data provided in Supplementary Tables 1 and 2. The recombinant Flp-In T-REx293-FLVCR1 and Flp-In T-REx293-FLVCR2 cell lines were generated by using a tetracycline-inducible and commercially available Flp-In T-REx293 host-cell line system from Invitrogen. Flp-In T-REx293 cells were cultured in high-glucose Dulbecco's Modified Eagle's Medium (DMEM; Sigma-Aldrich) supplemented with 10% fetal bovine serum (FBS; Gibco), 1% Pen/Strep (Gibco), 1 µg ml$^{-1}$ of Zeocin (Gibco) and 15 µg ml$^{-1}$ of blasticidin S hydrochloride (AppliChem) at 37 °C in an atmosphere of 5% $CO_2$. Cells were periodically tested negative for mycoplasma contamination. For stable integration, the pcDNA5/FRT-FLVCR1-FLAG and pcDNA5/FRT-FLVCR2-FLAG vectors were cotransfected with the Flp recombinase encoding expression vector pOG44 (Invitrogen) at a 1:13 mass ratio, respectively. All transfection procedures were performed with Lipofectamine 2000 reagent according to the manufacturer's instructions (Invitrogen). To select for stable clones, transfected cells were cultivated with growth medium containing 100 µg ml$^{-1}$ of hygromycin B (AppliChem).

## Transport assays in HEK293 cells

HEK293 cells were cotransfected with pcDNA3.1 plasmid and human *FLVCR1* or *FLVCR2* and human choline kinase A (*CHKA*) for choline transport assays or ethanolamine kinase 1 (*ETNK1*) for ethanolamine transport assays using Lipofectamine 2000 reagent (Invitrogen). Cells were periodically tested negative for mycoplasma contamination. After 24 h post-transfection, cells were incubated with DMEM containing 20 µM [$^3$H]choline or 2.5 µM [$^{14}$C]ethanolamine. The cells were incubated at 37 °C and 5% $CO_2$ for 1 h for uptake of the ligands. The cells were subsequently washed with ice-cold plain DMEM and lysed with RIPA buffer (Thermo Scientific) by shaking at room temperature for 30 min. The cell lysates were quantified by scintillation counter Tri-Carb (Perkin Elmer). Radioactive signals from cell lysates were normalized to total protein levels. For dose curve assays, indicated concentrations of choline and ethanolamine were incubated with the cells for 1 h at 37 °C. For time-course assays, the cells were incubated with 20 µM [$^3$H]choline or with 2.5 µM [$^{14}$C]ethanolamine. The transport assays were stopped at indicated time points by adding ice-cold plain DMEM. For testing transport activity of FLVCR1 mutants, 20 µM [$^3$H]choline and 2.5 µM [$^{14}$C]ethanolamine were used. For testing transport activity of FLVCR2 mutants, 100 µM [$^3$H]choline and 2.5 µM [$^{14}$C]ethanolamine were used. For transport assays of HEK293 cells overexpressing *FLVCR1* or *FLVCR2* without co-expressing with *CHKA* or *ETNK1*, 20 µM [$^3$H]choline and 2.5 µM [$^{14}$C]ethanolamine were used.

For transport assays under indicated pH conditions, the following buffers were used: pH 8.5 buffer (140 mM NaCl, 20 mM Tris-HCl pH 8.5, 2 mM $CaCl_2$, 1 g l$^{-1}$ of D-glucose), pH 6.5 buffer (140 mM NaCl, 20 mM MES pH 6.5, 2 mM $CaCl_2$, 1 g l$^{-1}$ of D-glucose) or pH 7.5 buffer (140 mM NaCl, 20 mM HEPES-NaOH, 2 mM $CaCl_2$, 1 g l$^{-1}$ of D-glucose). For sodium-free buffer, buffer containing 140 mM KCl, 20 mM HEPES-KOH pH 7.5, 2 mM $CaCl_2$, 1 g l$^{-1}$ of D-glucose was used. In these assays, 20 µM [$^3$H]choline or 2.5 µM [$^{14}$C]ethanolamine was used and the assays were stopped after 15 min of incubation with the ligands. Radioactive signals from cell lysates were normalized to total protein levels. Total protein was quantified using Pierce BCA Protein Assay Kit (Thermo Scientific).

## Immunofluorescent staining

HEK293 cells were seeded onto 24-well plates with coverslips and maintained in DMEM (Gibco) supplemented with 10% fetal bovine serum and 1% penicillin–streptomycin. HEK293 cells were cotransfected with *FLVCR1* or *FLVCR2* with membrane expressing GFP (Addgene: catalogue no. 14757) using Lipofectamine 2000 reagent (Invitrogen). The inducible HEK293 stable cell lines overproducing FLVCR1 or FLVCR2 were seeded onto Millicell EZ SLIDE eight-well glass slides (Millipore), respectively. The stable cell lines were induced at 80% confluence by adding a final concentration of 2 µg ml$^{-1}$ of doxycycline hydrochloride. The protein overproduction was carried out for 24 h. For permeabilization and staining, cells were washed with PBS twice and fixed in 4% PFA for 15 min at room temperature, followed by washing with PBS twice and permeabilized in PBST (PBS with 0.5% Triton-X) for 15 min at room temperature. For immunofluorescent staining, the HEK293 cells were subsequently washed with PBS and blocked in 5% normal goat serum for 1 h before staining with FLVCR1 and FLVCR2 polyclonal antibodies at 1:250 dilutions for 1 h and then with Alexa Fluor 555 (A-21428, Invitrogen) as secondary antibody at 1:500 dilutions for 1 h. The cells were counter-stained with DAPI (Thermo Scientific) and imaged with a laser confocal microscope (Zeiss LSM710). The overproduction stable cells were treated with the same protocol but stained with monoclonal ANTI-FLAG M2-FITC (F4049, Sigma-Aldrich) at 10 µg ml$^{-1}$ in TBS at room temperature for 1 h against their FLAG-tags and MitoTracker Red CMXRos (M7512, Invitrogen) for mitochondria localization. The cells were mounted using ProLong Diamond mounting medium with DAPI (P36966, Invitrogen) and imaged with laser confocal microscope (Confocal Microscope Leica STELLARIS 5).

## Structure-guided mutagenesis

To generate the mutant plasmids for *FLVCR1* and *FLVCR2*, an overlapping PCR approach was used. The mutated cDNA of *FLVCR1* or *FLVCR2* was cloned into pcDNA3.1 for overexpression. The mutations were validated by Sanger sequencing. To test the transport activity of these mutants, the mutant plasmid was either cotransfected with *CHKA* for choline transport assay or *ETNK1* for ethanolamine transport assay. After 24 h of post-transfections, cells were washed with DMEM and incubated with DMEM containing 20 µM [$^3$H]choline or 2.5 µM [$^{14}$C]ethanolamine for *FLVCR1* mutants and 100 µM [$^3$H]choline or 2.5 µM [$^{14}$C]ethanolamine for *FLVCR2* mutants. The assays were stopped after 1 h of incubation at 37 °C. Radioactive signal of each mutant was normalized to the total protein levels.

## Choline export assay

To examine the export function, *FLVCR1* and *FLVCR2* plasmids were expressed in HEK293 cells without cotransfection with *CHKA* or *ETNK1*. The cells were then incubated with 200 µM [$^3$H]choline or 100 µM [$^{14}$C] ethanolamine for 2 h to prepack the cells with the ligand. Subsequently, the cells were washed to remove the ligands left over in the medium and incubated with choline/ethanolamine-free medium for 1 h at 37 °C for the release of prepacked ligand. The cells were washed and collected for quantification of radioactive signals. Samples after 2 h of incubation with the radioactive ligand were collected to determine the levels of radioactive levels before the release and used for control.

## Metabolomic analysis

Adult livers (aged 3–6 months) from controls (*FLVCR1*$^{f/f}$ and *FLVCR1*$^{f/+}$-Mx1-Cre) and conditional *FLVCR1*-knockout (*FLVCR1*$^{f/f}$-Mx1-Cre) mice were used for metabolomic analysis. All mice were maintained under specific pathogen-free conditions with free access to food and water with 12 h dark–light cycle. Briefly, the mice were perfused with PBS to remove blood before organ collection. Liver samples were snap-frozen before being shipped for metabolomics by Metabolon. The levels of metabolites were expressed as relative amount. Studies involving mice were reviewed and approved by the University of Washington Institutional Animal Care and Use Committee under protocol number 2001-13.

## Production and purification of the human FLVCR1 and FLVCR2

For protein production, the Flp-In T-REx293-FLVCR1 and Flp-In T-REx293-FLVCR2 cell lines were cultured in roller bottles (Greiner Bio-One) in growth media containing 100 µg ml$^{-1}$ of hygromycin B for 14 d under the above-mentioned conditions. Gene expression was induced at 100% confluence by adding a final concentration of 2 µg ml$^{-1}$ of doxycycline hydrochloride. After 72 h, cells were harvested with Accutase solution (Sigma-Aldrich) and stored at −80 °C until further use. Harvested cells were suspended in cold lysis buffer containing 25 mM Tris pH 7.4, 150 mM NaCl and 0.1 g ml$^{-1}$ of SigmaFast ethylenediaminetetraacetic acid (EDTA)-free protease inhibitor (Sigma-Aldrich) and disrupted by stirring under high-pressure nitrogen atmosphere (750 MPa) for 45 min at 4 °C in a cell-disruption vessel (Parr Instrument). The cell lysate was centrifuged at 8,000g at 4 °C for 15 min. Subsequently, the low-velocity supernatant was centrifuged at 220,000g at 4 °C for 60 min. Pelleted membranes were resuspended and stored in a storage buffer containing 25 mM Tris pH 7.4, 150 mM NaCl, 10% glycerol (v/v) and 0.1 g ml$^{-1}$ of SigmaFast EDTA-free protease inhibitor (Sigma-Aldrich).

All purification steps of both FLVCRs were performed at 4 °C. Isolated membranes were solubilized with 1% (w/v) lauryl maltose neopentyl glycol (LMNG; GLYCON Biochemicals) with gentle stirring for 1 h. The insoluble membrane fraction was removed through ultracentrifugation at 220,000g for 1 h. Subsequently, the supernatant was incubated with ANTI-FLAG M2 Affinity Gel resin (Millipore) for 1 h. The resin was pre-equilibrated with a buffer containing 50 mM Tris pH 7.4, 150 mM NaCl and 0.02% LMNG (w/v). The washing step was performed using 20 column volumes of wash buffer (50 mM Tris pH 7.4, 150 mM NaCl, 5% (v/v) glycerol and 0.02% LMNG). The protein was eluted from the M2 resin with 10 column volumes of the same buffer supplemented with 4 mM FLAG Peptide (Millipore). The eluted sample was concentrated and subjected to a Superdex 200 Increase 10/300 column (Cytiva) equilibrated with size exclusion chromatography buffer (50 mM Tris pH 7.4, 150 mM NaCl and 0.001% (w/v) LMNG). Peak fractions were pooled, concentrated to 1.5 mg ml$^{-1}$ using an Amicon 50 kDa cut-off concentrator (Millipore) and stored for further analysis.

## Immunoblotting

Affinity-purified proteins were subjected to SDS–polyacrylamide gel electrophoresis and immunoblotting. FLAG-tagged FLVCR1 and FLVCR2 were detected using anti-FLAG (F3165, Sigma-Aldrich) at 1:1,000 dilution. Anti-mouse IgG conjugated with alkaline phosphatase (A9316, Sigma-Aldrich) was used as secondary antibody at 1:5,000 dilution. Native FLVCR1 and FLVCR2 proteins were detected by polyclonal FLVCR1 and FLVCR2 antibodies raised in-house at 1:1,000 dilution. GAPDH antibody (sc-32233, Santa Cruz) was used as loading control at 1:4,000 dilution. IRDye 680LT (926-32212, Li-COR Biosciences) was used as secondary antibody for detection.

## Tryptophan fluorescence measurement

Tryptophan fluorescence measurements were carried out using Prometheus Panta (NanoTemper Technologies). Purified protein samples were diluted with dilution buffer containing 50 mM HEPES pH 7.4, 150 mM NaCl and 0.001% (w/v) LMNG to 1 µM. Buffers with different concentrations of choline or betaine were prepared by serial dilutions in dilution buffer containing 4 mM of the compounds. The protein samples were mixed with an equal volume of dilution buffer or the compound-containing buffer with a final protein concentration of 0.5 µM and then incubated at room temperature for 15 min. A volume of 10 µl of mixed solution was used per Prometheus high-sensitivity capillary (NanoTemper Technologies). Recorded $F_{350}/F_{330}$ was analysed by using Python libraries including pandas, numpy, scipy and seaborn in Visual Studio Code (Microsoft). Three technical replicates were recorded for data analysis. The custom python code used for data analysis is publicly available through https://doi.org/10.5281/zenodo.10938397.

## Cryo-EM sample preparation

To collect cryo-EM data of FLVCR1 and FLVCR2 in different sample conditions, different combinations of FLVCR proteins and putative substrate molecules were prepared. For both as-isolated samples of FLVCRs, the protein concentration was adjusted to approximately 1.5 mg ml$^{-1}$ and subjected to plunge freezing. For samples supplemented with choline, purified proteins were adjusted to 1.5 mg ml$^{-1}$ and choline was added at a final concentration of 1 mM. For FLVCR1 samples supplemented with ethanolamine, purified proteins were adjusted to 1.5 mg ml$^{-1}$ and ethanolamine was added at a final concentration of 1 mM. The samples were incubated for 10 min at room temperature before plunge freezing. Identical plunge freezing conditions were applied for all samples: 300 mesh R1.2/1.3 copper grids (Quantifoil) were washed in chloroform and subsequently glow-discharged with a PELCO easiGlow device at 15 mA for 90 s. A volume of 4 µl sample was applied to a grid and blotting was performed for 4 s at 4 °C, 100% relative humidity with nominal blot force 20 immediately before freezing in liquid ethane, using a Vitrobot Mark IV device (Thermo Scientific).

## Cryo-EM image recording

For each cryo-EM sample, a dataset was recorded in energy-filtered transmission electron microscopy mode using either a Titan Krios G3i or a Krios G4 microscope (Thermo Scientific), both operated at 300 kV. Electron-optical alignments were adjusted with EPU software 3.0–3.4 (Thermo Scientific). Images were recorded using automation strategies of EPU 3.0–3.4 in electron counting mode with either a Gatan K3 (installed on Krios G3i) or a Falcon4 (installed on Krios G4) direct electron detector. For Gatan K3 detector, a nominal magnification of 105,000, corresponding to a calibrated pixel size of 0.837 Å was used and dose fractionated videos (80 frames) were recorded at an electron flux of approximately 15 e$^-$ pixel$^{-1}$ s$^{-1}$ for 4 s, corresponding to a total dose of about 80 e$^-$ A$^{-2}$. For Falcon4 detector, a nominal magnification of 215,000, corresponding to a calibrated pixel size 0.573 Å was used, dose fractionated videos were recorded in electron-event representation format at an electron flux of approximately 4 e$^-$ pixel$^{-1}$ s$^{-1}$ for 5 s, corresponding to a total dose of about 50 e$^-$ A$^{-2}$. Images were recorded between −1.1 and −2.0 µm nominal defocus. Data collection quality was monitored through EPU v.3.0-3.4 and CryoSparc Live (v.3.0 and 4.0)[33].

## Cryo-EM image processing

For each acquired dataset, the same cryo-EM image processing approach was applied: MotionCor2 was used to correct for beam-induced motion and to generate dose-weighted images[34]. Gctf was used to determine the contrast transfer function (CTF) parameters and perform correction steps[35]. Images with estimated poor resolution (more than 4 Å) and severe astigmatism (more than 400 Å) were removed at this step. Particles were picked by TOPAZ and used for all further processing steps[36]. Two-dimensional classification, initial model generation, three-dimensional (3D) classification, CTF refinement, Bayesian polishing, 3D sorting and final map reconstructions were performed using RELION (v.3.1 and 4.0) or cryoSPARC (v.3.0 and 4.0)[33,37,38]. In the data processing pipeline, 3D autorefine jobs were conducted following each 3D classification or 3D sorting round for all resulted classes, to carefully assess the resulting density maps for quality and resolution through both metrics and visual inspection. Data processing was only proceeded with those maps that seemed promising for further refinement stages. Fourier shell correlation (FSC) curves and local-resolution estimation were generated in RELION or cryoSPARC for individual final maps. A schematic overview of our processing workflow and a summary of map qualities are shown and Supplementary Figs. 3–5.

## Model building and geometry refinement

The first atomic models of FLVCR1 and FLVCR2 were built into the respective electron microscopy density maps of the as-isolated state in Coot (v0.8) or ISOLDE within ChimeraX (v.1.5 and 1.6)[39–41], using the AlphaFold predicted structures as initial templates[42]. After manual backbone tracing and docking of side chains, real-space refinement in Phenix was performed (v.1.18)[43]. Refinement results were manually inspected and corrected if required. These models were used as templates to build all subsequent atomic models. The finalized models were validated by MolProbity implemented in Phenix[44]. Map-to-model cross-validation was performed in Phenix (v.1.18). $FSC_{0.5}$ was used as cut-off to define resolution. The comprehensive information on the Cryo-EM data collection, refinement and validation statistics is shown in Extended Data Table 1. The finalized models of the two FLVCR proteins in different states were visualized using ChimeraX and used as starting structures for molecular dynamics simulations.

## Molecular dynamics simulations

All molecular dynamics simulations were performed using the GROMACS 2022.4 (ref. 45) software. The protein structures were embedded in a lipid bilayer with 75% POPE and 25% POPG with CHARMM-GUI[46] and solvated in TIP3P water with 150 mM NaCl. The CHARMM36m force field[47] was used with the improved WYF parameters for cation–π interactions, in particular of the choline and ethanolamine ligands[48]. The systems were minimized for 5,000 steepest-descent steps and equilibrated for 250 ps of molecular dynamics in an NVT ensemble and for 1.625 ns in an NPT ensemble. Position restraints of 4,000 and 2,000 kJ mol$^{-1}$ nm$^{-2}$ in the backbone and side chain heavy atoms, respectively, were gradually released during equilibration. The $z$-positions of membrane phosphates, as well as lipid dihedrals, were initially restrained with force constants of 1,000 kJ mol$^{-1}$ nm$^{-2}$, which were gradually released during equilibration. The initial time step of 1 fs was increased to 2 fs during NPT equilibration. Long-range electrostatic interactions were treated with particle-mesh Ewald[49] with a real-space cut-off of 1.2 nm. Van-der-Waals interactions were cut-off beyond a distance of 1.2 nm. The LINCS algorithm[50] was used to constrain the bonds involving hydrogen atoms. During equilibration, a constant temperature of 310 K was maintained with the Berendsen thermostat[51], using a coupling constant of 1 ps. Constant pressure of 1 bar was established with a semi-isotropic Berendsen barostat and a coupling constant of 5 ps. In the production runs, a Nosé–Hoover thermostat[52] and a Parrinello–Rahman barostat were used[53].

We used our cryo-EM structures as initial models for simulations of as-isolated and choline-bound inward-facing FLVCR1, as-isolated and choline-bound inward-facing FLVCR2 and as-isolated outward-facing FLVCR2. We set the protonation state of each residue as predicted for pH 7.0 using the PROPKA server[54]. An initial structure of choline-bound outward-facing FLVCR2 was generated by aligning the as-isolated outward-facing FLVCR2 to the choline-bound inward-facing FLVCR2 and maintaining choline in the cavity. In choline entry simulations, the as-isolated structures were used with 380 mM choline in solution. For simulations of ethanolamine-bound FLVCR1, the choline within the cavity of the cryo-EM structure was replaced by this ligand. Simulations with deprotonated ethanolamine were performed as well and results are included in Supplementary Figs. 11 and 12. Choline and ethanolamine release simulations were interrupted after ligand exit from the cavity and hence have variable duration. For all other systems, each replica was run for 1 μs. A summary of all simulations performed in this study is provided in Supplementary Table 3 (table with technical information). Time-resolved distance calculations for all replicas not included in the main figures are shown in Supplementary Fig. 11. Minimum atom-pair distances were calculated as the minimum distance over all pairs of atoms in two stated groups, including hydrogens (for example, ligand and certain defined side chains).

Alanine substitution mutations were introduced using PyMol[55] and simulated with identical parameters as those applied in the corresponding wild-type simulations. In FLVCR1 and FLVCR2, alanine mutations were introduced in the cavity residues W125[FLVCR1] and W102[FLVCR2], respectively.

For the MM/PBSA calculations, we used gmx_MMPBSA[56] with dielectric constants of 7.0, 80.0 and 4.0 for the membrane, solvent and protein, respectively, and the default surface tension of 0.0072 kcal mol$^{-1}$ nm$^{-2}$. We estimated entropies using the interaction entropy method[57]. The contributions of W125[FLVCR1] and W102[FLVCR2] to the binding energy were estimated by alanine scanning.

Visual molecular dynamics[58] and MDAnalysis[59] were used to visualize and analyse the trajectories, respectively. An assessment of the reliability and reproducibility of our simulations is provided in Supplementary Table 4.

## Interior tunnels and cavities

Tunnels and cavities were mapped with MOLE v.2.5 (ref. 60) with a bottleneck radius of 1.2 Å, bottleneck tolerance 3 Å, origin radius 5 Å, surface radius 10 Å, probe radius 5 Å and an interior threshold of 1.1 Å.

We calculated the volume of the cavity using CASTp[61] with a bottleneck radius of 1.4 Å. Residues 297–320 and 512–516 were removed from the FLVCR1 model to avoid the misattribution of the volume between internal loops to the cavity volume. Analogously, residues 272–296 and 487–502 were not included in the cavity volume calculation of FLVCR2.

## Sequence alignments

Multiple sequence alignments of FLVCR1 and FLVCR2 from *Homo sapiens*, *Felis catus*, *Mus musculus* and *Sus scrofa* were performed using Clustal Omega[62].

## Reporting summary

Further information on research design is available in the Nature Portfolio Reporting Summary linked to this article.

## Data availability

Cryo-EM maps are deposited to the Electron Microscopy Data Bank under accession numbers: EMD-18334, EMD-18335, EMD-18336, EMD-18337, EMD-18339 and EMD-19009. Atomic models of human FLVCR1 and FLVCR2 have been deposited to the Protein Data Bank under accession numbers: 8QCS, 8QCT, 8QCX, 8QCY, 8QD0 and 8R8T. All molecular dynamics trajectories generated for this study and simulation input files are deposited in a Zenodo repository and freely available at https://doi.org/10.5281/zenodo.10952971 (ref. 63). All other data are presented in the main text or Supplementary Information. Source data are provided with this paper.

## Code availability

The custom python code used for the analysis of tryptophan fluorescence data is publicly available in a Zenodo repository at https://doi.org/10.5281/zenodo.10938397 (ref. 64).

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

**Acknowledgements** We thank H. Michel for supporting and providing infrastructural resources. We thank the Central Electron Microscopy Facility at Max Planck Institute of Biophysics for technical support and access to instrumentation. We thank A. Kannt (Fraunhofer Leistungszentrum TheraNova) and G. Geißlinger (Fraunhofer ITMP, Goethe University Frankfurt) for support and helpful discussions. This work was supported by the Max Planck Society and the Nobel Laureate Fellowship of the Max Planck Society (to D.W. and S.S.) and Singapore Ministry of Education grant nos. T2EP30221-0012, T2EP30123-0014 and NUHSRO/2022/067/T1 (to L.N.N.).

**Author contributions** T.-H.W. purified proteins, performed biochemical assays, prepared grids, collected cryo-EM data, processed cryo-EM data, refined the structure, built the model, codrafted the manuscript and prepared figures. K.R., Y.Z. and T.J.Y.L. performed mutagenesis. K.R. performed transport assays and western blot analyses for mutants and export assays. N.C.P.L. and Y.Z. performed dose curve and time-course assays and immunostaining. A.C.C. performed molecular dynamics simulations, analysed data, codrafted the manuscript and prepared figures. W.J. performed cell productions, optimized purification conditions and purified proteins. A.B. performed assays and analysed data. R.T.D. and J.L.A. designed and performed animal model experiments. S.W. calibrated and aligned the microscopes. G.G. performed immunofluorescence assays and protein purification. S.L.S. performed protein purification. G.H., D.W., L.N.N. and S.S. supervised the project. D.W. implemented cell production and protein purification, prepared grids, performed initial cryo-EM screening experiments, collected cryo-EM data, analysed data, drafted the manuscript, generated figures and funded the project. L.N.N. designed functional assays, interpreted and analysed data, funded the project and codrafted the manuscript. D.W. and S.S. initiated the project. S.S. designed research, evaluated data, funded the project, drafted the manuscript and generated figures.

**Funding** Open access funding provided by Max Planck Society.

**Competing interests** The authors declare no competing interests.

**Additional information**
**Correspondence and requests for materials** should be addressed to Gerhard Hummer, Di Wu, Long N. Nguyen or Schara Safarian.

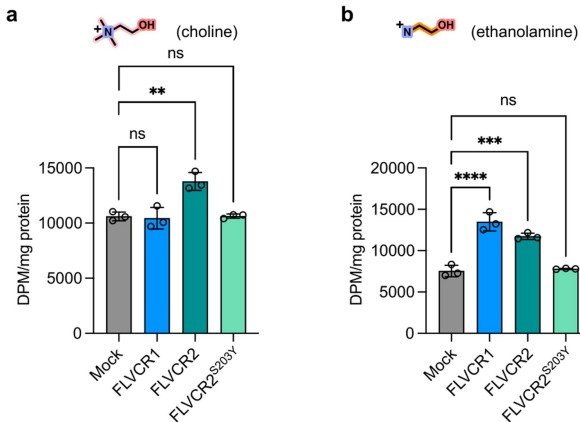

**Extended Data Fig. 1 | Transport activity of FLVCRs for choline or ethanolamine without overexpression of *CHKA* or *ETNK-1*.** Choline (**a**) or ethanolamine (**b**) transport activity of FLVCR1, FLVCR2 and FLVCR2$^{S203Y}$ without simultaneous overexpression of *CHKA* or *ETNK-1*, respectively. 20 μM [$^{3}$H] choline or 2.5 μM [$^{14}$C] ethanolamine was used, respectively. The inactive mutant of FLVCR2$^{S203Y}$ was used as a control in all experiments. Each experiment was repeated twice and one dataset was shown. n = 3 biologically independent samples (wells). Data are expressed as mean ± SD. In (**a**) **P = 0.002. In (**b**) ****P < 0.0001, ***P = 0.0003. One-way ANOVA.

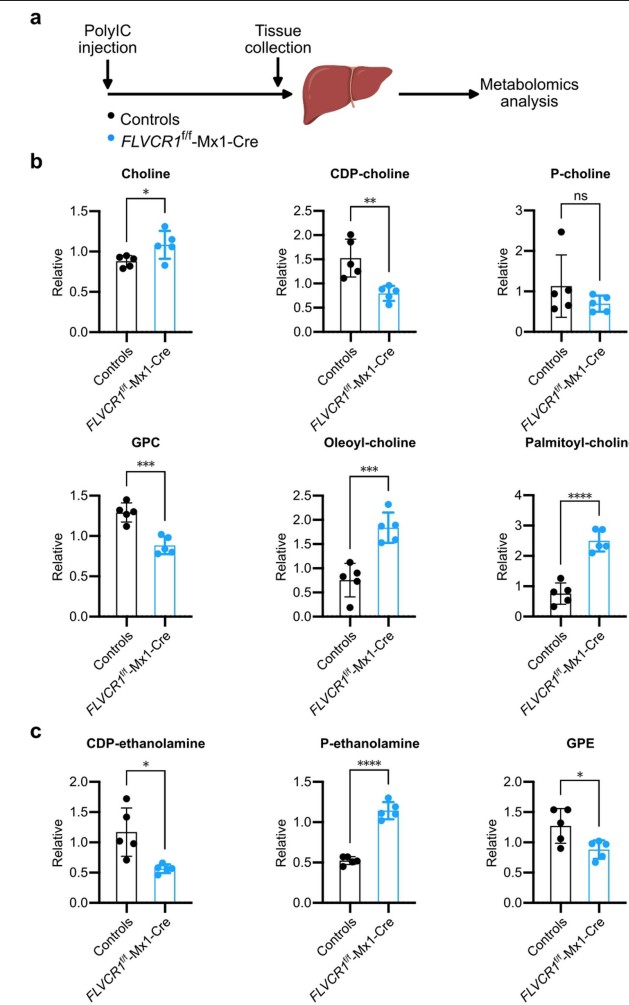

**Extended Data Fig. 2 | Metabolomic analysis of livers from *FLVCR1*-knockout mice. a**, Illustration of experimental procedures. *FLVCR1* was deleted by polyIC injection in *FLVCR1*^f/f^-Mx1-Cre (knockout) mice. Liver samples of controls (*FLVCR1*^f/f^ and *FLVCR1*^f/+^-Mx1-Cre) and knockout mice were collected at least 4 weeks post-injection for metabolomic analysis. The illustration was created with BioRender.com. **b**, Levels of choline and choline metabolites from controls and knockout mice. **c**, Levels of ethanolamine metabolites from controls and knockout mice. GPC, glycerophosphocholine; GPE, glycerophosphoethanolamine. Each data point represents one mouse. n = 5 independent mice. Data are expressed as mean ± SD. In (**b**) *P = 0.0411, **P = 0.0047, ***P = 0.0005, ***P = 0.0008, ****P < 0.0001 from left to right and from top row to bottom row, respectively; In (**c**) *P = 0.0103, ****P < 0.0001, *P = 0.0279 from left to right, respectively. ns, not significant. Unpaired two-tailed *t*-test for (**b**) and (**c**).

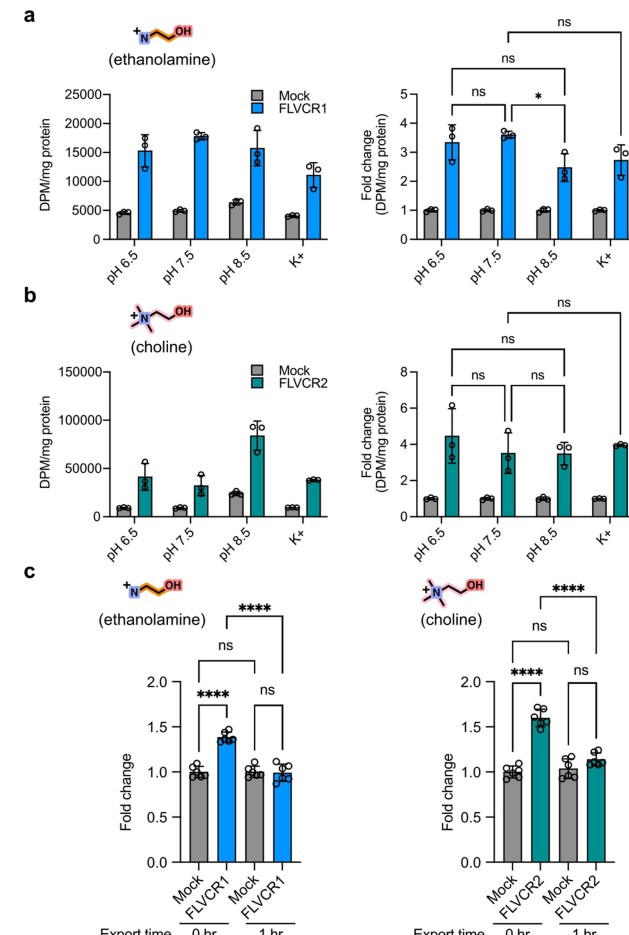

**Extended Data Fig. 3 | Transport properties of FLVCR1 and FLVCR2.**
**a**, Ethanolamine transport activity of FLVCR1 at indicated pH values and in sodium-free buffer (K+). **b**, Choline transport activity of FLVCR2 at indicated pH values and in sodium-free buffer (K+). The right panels show normalized data with reference to the respective mocks. In these experiments, 2.5 μM [14C] ethanolamine was used for FLVCR1 or 20 μM [3H] choline for FLVCR2, respectively. The cells were co-expressed with *ETNK-1* or *CHKA* and incubated with the ligands at 37 °C for 15 mins. Experiments were repeated twice on different days. A second dataset for (**a**) and (**b**) was also provided in the source data file. **c**, Export assays of FLVCR1 with ethanolamine (left) and FLVCR2 with choline (right). In these assays, 100 μM [14C] ethanolamine or 200 μM [3H] choline was incubated with *FLVCR1* or *FLVCR2* overexpression cells, respectively. After 2 h of incubation, the buffer was washed out. Intracellular [3H] choline or [14C] ethanolamine from the cells was allowed to release into choline/ ethanolamine-free medium for 1 h. The radioactive signal in the cells was normalized to the total protein and expressed as fold change to mock. n = 3 (**a**,**b**) and n = 6 (**c**) biologically independent samples (wells). Data are expressed as mean ± SD. In (**a**) *P = 0.0222. In (**c**) ****P < 0.0001. ns, not significant. Two-way ANOVA for (**a**) and (**b**), One-way ANOVA for (**c**).

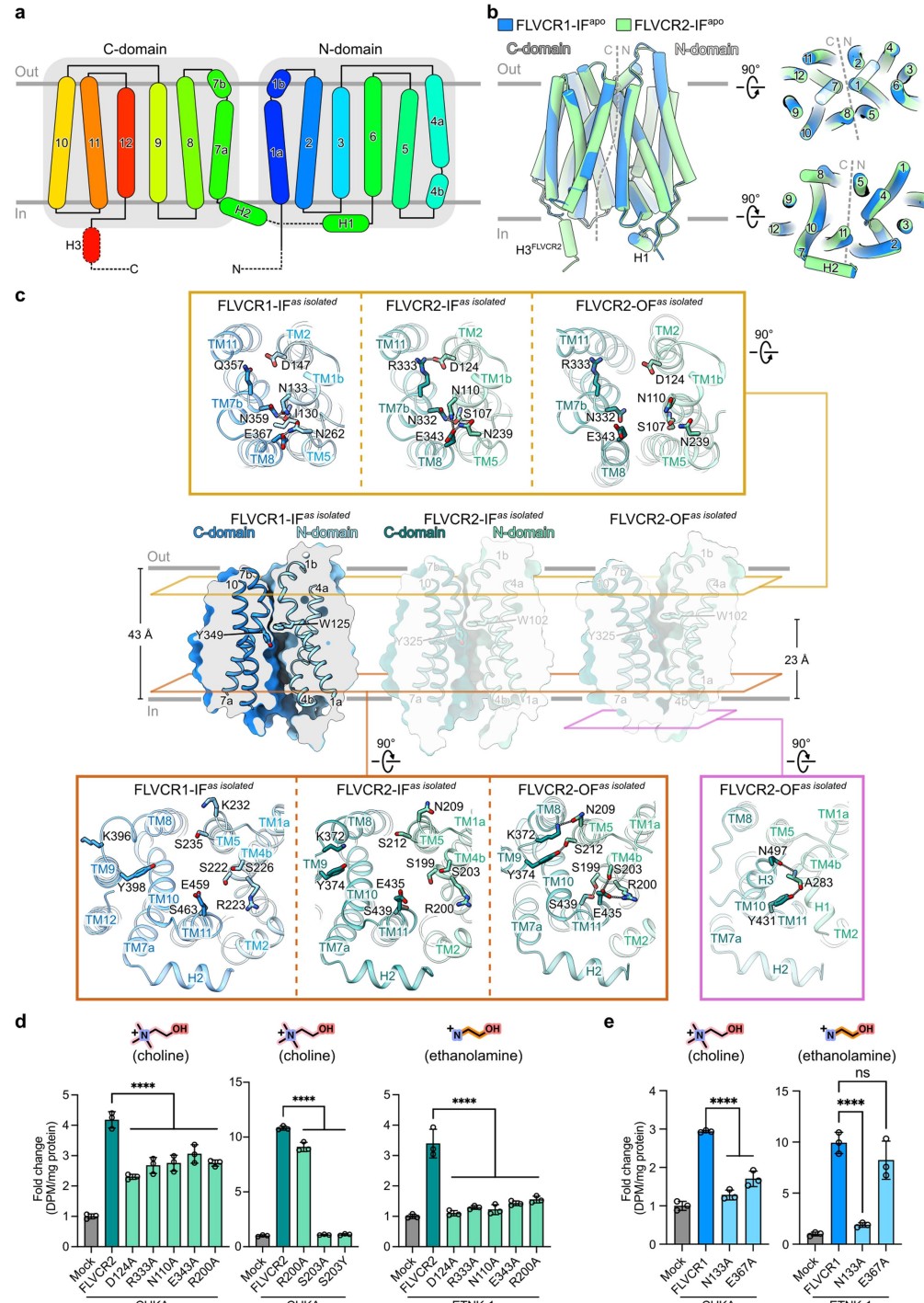

**Extended Data Fig. 4 | Overall architecture and gating residues of FLVCR1 and FLVCR2. a**, Schematic diagram of FLVCR family showing the topology of the secondary structure. Motifs that are not observed in both cryo-EM structures of FLVCR1 and FLVCR2 are shown as dashed lines. **b**, Structural comparison of FLVCR1-IF$^{as\,isolated}$ and FLVCR2-IF$^{as\,isolated}$ in tube representation, viewed from the lipid bilayer (left), extracellular side (top-right) and intracellular side (bottom-right). **c**, Cut-away view of FLVCR1-IF$^{as\,isolated}$ in surface representation showing the cytoplasmic cavity. Two central aromatic residues are shown as sticks. The structures of FLVCR2-IF$^{as\,isolated}$ and FLVCR2-OF$^{as\,isolated}$ are shown for comparison. Cross-sections of their interdomain interactions are shown from the extracellular side (top), or from the intracellular side (bottom-left). Residues in FLVCR1 corresponding to the interdomain interaction residues in FLVCR2 are shown. The bottom-right panel shows the interdomain interactions between

H1 and H3 in FLVCR2-OF$^{as\,isolated}$ viewed from the intracellular side. Residues participating in the interdomain interactions are shown as sticks; hydrogen bonds and salt bridges are labelled with dashed lines. **d**, Transport assay of FLVCR2 mutants for choline (left) and ethanolamine (right). *CHKA* or *ETNK-1* was co-expressed with wild-type *FLVCR2* and mutant plasmids. In these assays, 100 µM [³H] choline or 2.5 µM [¹⁴C] ethanolamine was used, respectively. **e**, Transport assay of FLVCR1 mutants for choline (left) and ethanolamine (right). *CHKA* or *ETNK-1* was co-expressed with wild-type *FLVCR1* and mutant plasmids and 20 µM [³H] choline or 2.5 µM [¹⁴C] ethanolamine was used, respectively. Separate experiments for each panel were performed in (**d**) and (**e**). n = 3 biologically independent samples (wells). Data shown are mean ± SD. In (**d**) ****P < 0.0001. In (**e**) ****P < 0.0001. ns, not significant. One-way ANOVA for (**d**) and (**e**).

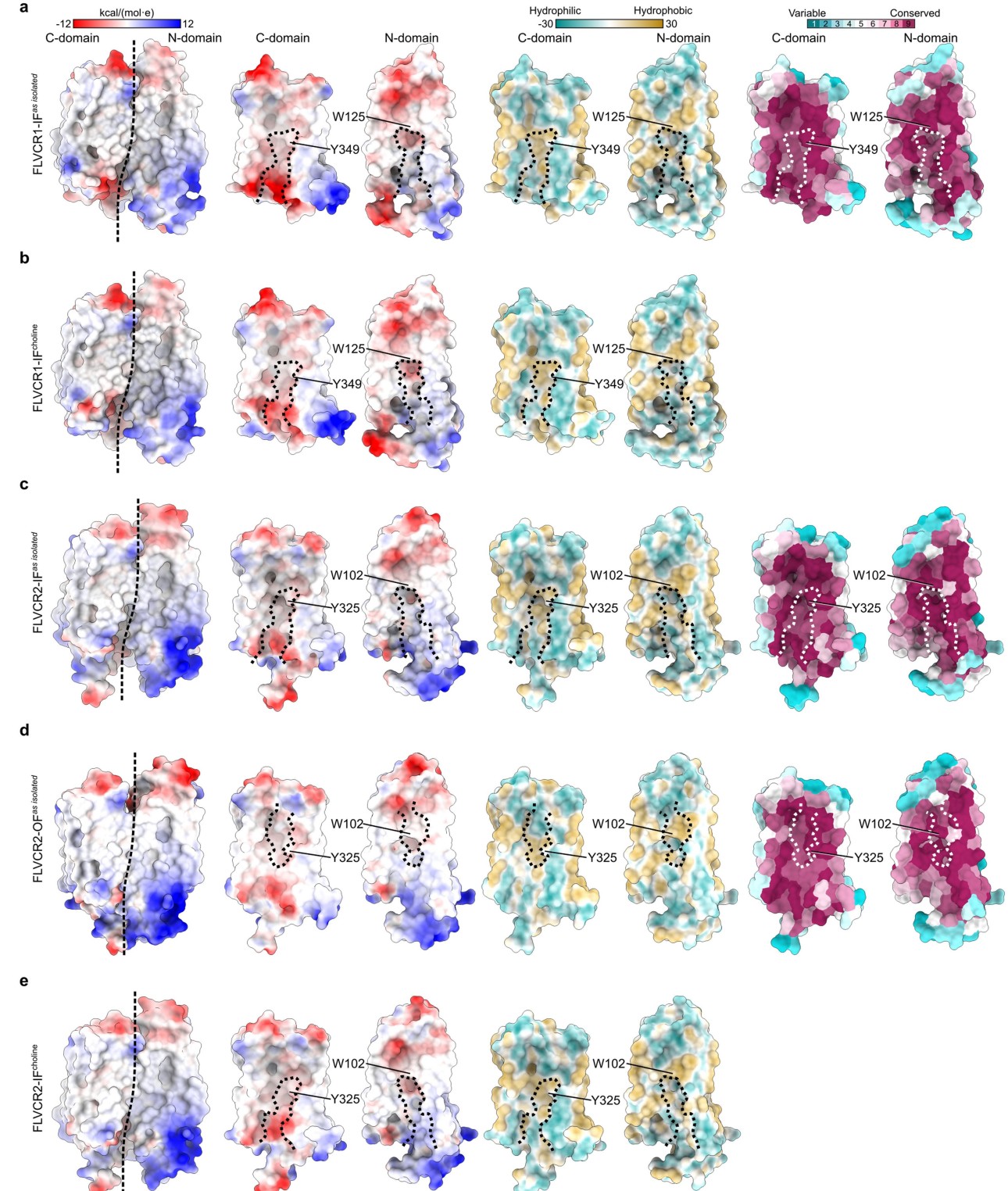

**Extended Data Fig. 5 | Physicochemical properties of FLVCR structures and conservation analyses.** Surface charge, hydrophobicity and conservation analyses of FLVCR1-IF$^{as\,isolated}$ (**a**), FLVCR1-IF$^{choline}$ (**b**), FLVCR2-IF$^{as\,isolated}$ (**c**), FLVCR2-OF$^{as\,isolated}$ (**d**) and FLVCR2-IF$^{choline}$ (**e**). From left to right: Surface viewed from the lipid bilayer and both C and N domains viewed from the domain interface coloured by electrostatic potential, the domain interface coloured by hydrophobicity and the domain interface coloured by sequence conservation. The cavity in the different states of both FLVCRs is outlined with dashed lines. Two conserved central pocket residues (W125 and Y349 in FLVCR1; W102 and Y325 in FLVCR2) are indicated. Conservation analysis was performed using the ConSurf server (https://consurf.tau.ac.il/).

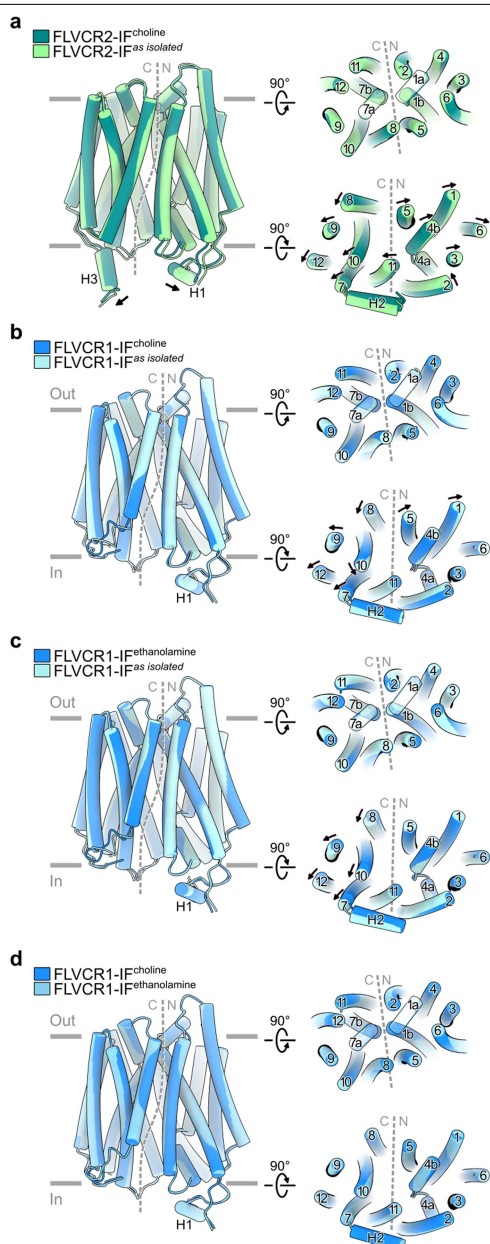

**Extended Data Fig. 6 | Conformational changes of substrate-bound FLVCRs.** Structural comparison of FLVCR2-IF^choline and FLVCR2-IF^as isolated (**a**), FLVCR1-IF^choline and FLVCR1-IF^as isolated (**b**), FLVCR1-IF^ethanolamine and FLVCR1-IF^as isolated (**c**), as well as FLVCR1-IF^choline and FLVCR1-IF^ethanolamine (**d**), viewed from the lipid bilayer (left), extracellular side (top-right) and intracellular side (bottom-right).

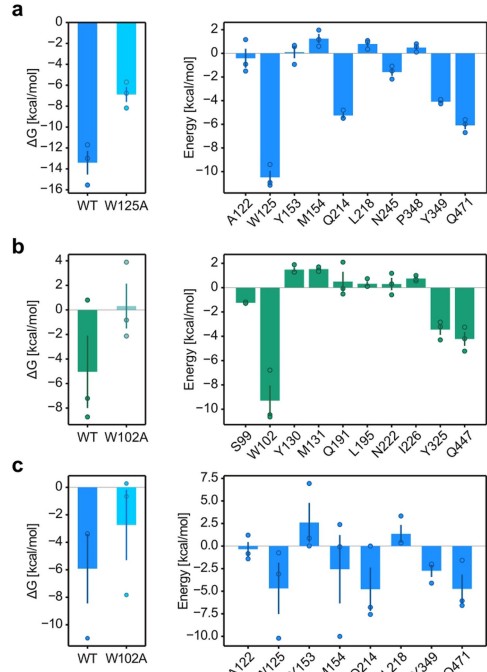

**Extended Data Fig. 7 | Binding free energy contributions of ligand-coordinating residues in FLVCR1 and FLVCR2. a,b**, MM/PBSA free energy calculations for choline in the binding pocket of inward-facing FLVCR1 (**a**) and inward-facing FLVCR2 (**b**). **c**, MM/PBSA free energy calculations for ethanolamine in the binding pocket of FLVCR1. Alanine scanning mutations were conducted on residues W125[FLVCR1] and W102[FLVCR2] in order to quantitatively assess their respective contributions (left). Energy decomposition was carried out for all residues within a 4 Å radius of choline (right). Bars represent the mean values obtained from three independent replicas (points), with error bars representing s.e.m. The last 200 ns of replica 1 of FLVCR2 were not included in the free energy calculations due to choline release from the binding site. Analogously, free energy calculations for ethanolamine were performed only for the frames in which the ligand remained in the cavity (375 ns in total).

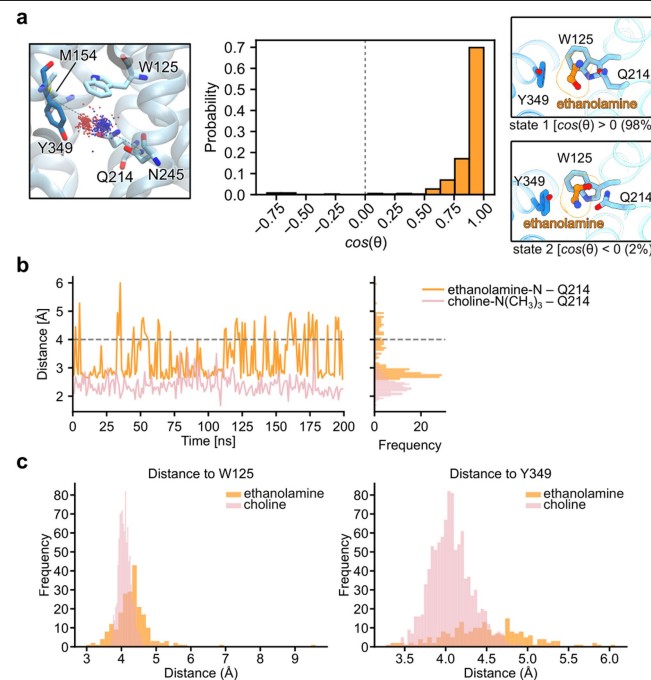

**Extended Data Fig. 8 | Interactions and dynamics of ethanolamine binding to inward-facing FLVCR1. a**, In molecular dynamics simulations, ethanolamine binds to inward-facing FLVCR1 in two distinct orientations, as quantified by the histogram of the cosine of the angle $\theta$ between the vectors connecting the N and O atoms of ethanolamine and the Cα atoms of N245[FLVCR1] and M154[FLVCR1]. The primary amine (blue dots in the zoom-in) remains sandwiched by the aromatic side chains and the OH (red dots in the zoom-in) stays close to Q214[FLVCR1], whose amide also moves and flips. **b**, Distance between Q214[FLVCR1] and the primary amine of ethanolamine (orange) and, for reference, the tertiary amine of choline (pink) as function of time in molecular dynamics simulations, with distance distributions on the right. **c**, Distribution of distances between the N-atom of ethanolamine or choline and the highly conserved tryptophan and tyrosine residues in molecular dynamics simulations of FLVCR1.

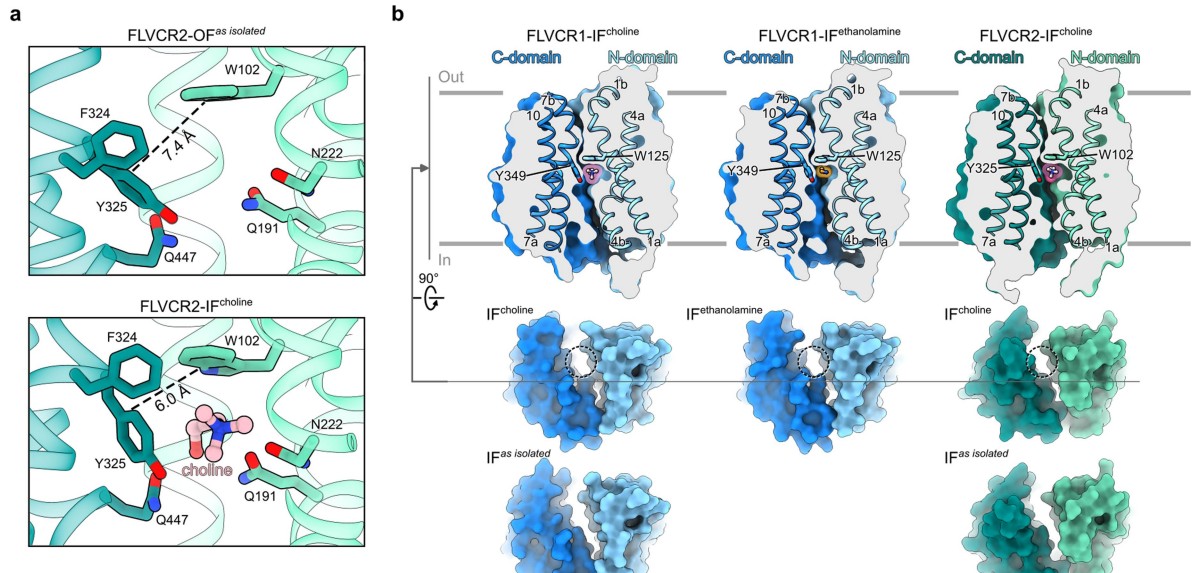

**Extended Data Fig. 9 | Translocation pathway of choline in FLVCRs. a**, Choline-binding sites of FLVCR2-OF$^{as\,isolated}$ (top) and FLVCR2-IF$^{choline}$ (bottom) with the distance between W102 and Y325 shown as dashed lines. **b**, Cut-away views of FLVCR1-IF$^{choline}$ (top-left), FLVCR1-IF$^{ethanolamine}$ (top-middle) and FLVCR2-IF$^{choline}$ (top-right) showing the inward-facing cavity. Two central aromatic residues are shown as sticks. The surfaces shown below are the respective models viewed from the intracellular side. The surfaces of IF$^{as\,isolated}$ of FLVCR1 and FLVCR2 are also shown for comparison. The dashed circles indicate the peripheral channel in the ligand-bound state.

## Extended Data Table 1 | Cryo-EM Data collection, refinement and validation statistics

| | FLVCR1-IF*as isolated* (EMD-18334; PDB 8QCS) | FLVCR1-IF*choline* (EMD-18335; PDB 8QCT) | FLVCR1-IF*ethanolamine* (EMD-19009; PDB 8R8T) | FLVCR2-IF*as isolated* (EMD-18336; PDB 8QCX) | FLVCR2-OF*as isolated* (EMD-18337; PDB 8QCY) | FLVCR2-IF*choline* (EMD-18339; PDB 8QD0) |
|---|---|---|---|---|---|---|
| **Data collection and processing** | | | | | | |
| Magnification | 215,000 | 215,000 | 215,000 | 215,000 | 215,000 | 215,000 |
| Voltage (kV) | 300 | 300 | 300 | 300 | 300 | 300 |
| Electron dose (e⁻/Å²) | 55 | 55 | 55 | 80 | 80 | 57 |
| Defocus range (μm) | −1.1 to −2.1 | −1.1 to −2.1 | −1.1 to −2.1 | −1.1 to −2.1 | −1.1 to −2.1 | −1.1 to −2.1 |
| Pixel size (Å) | 0.573 | 0.573 | 0.573 | 0.573 | 0.573 | 0.573 |
| Symmetry imposed | $C1$ | $C1$ | $C1$ | $C1$ | $C1$ | $C1$ |
| Initial particle images (no.) | 3,247,307 | 3,251,081 | 5,185,344 | 3,497,517 | 3,497,517 | 4,790,154 |
| Final particle images (no.) | 56,664 | 25,873 | 52,147 | 50,167 | 49,532 | 51,511 |
| Map resolution (Å) | 2.9 | 2.6 | 2.9 | 3.1 | 2.9 | 2.8 |
| FSC threshold | 0.143 | 0.143 | 0.143 | 0.143 | 0.143 | 0.143 |
| Map resolution range (Å) | 2.4–6.5 | 2.6–4.9 | 2.8–5.0 | 2.8–5.5 | 2.7–3.8 | 2.6–3.8 |
| | | | | | | |
| **Refinement** | | | | | | |
| Initial model used | AlphaFold model (AF-Q9Y5Y0-F1) | | | AlphaFold model (AF-Q9UPI3-F1) | | |
| Model resolution (Å) | 3.1 | 2.6 | 2.9 | 3.4 | 3.3 | 3.0 |
| FSC threshold | 0.5 | 0.5 | 0.5 | 0.5 | 0.5 | 0.5 |
| Model resolution range (Å) | | | | | | |
| Map sharpening $B$ factor (Å²) | −80 | −30 | −50 | −120 | −100 | −30 |
| Model composition | | | | | | |
| Non-hydrogen atoms | 3,241 | 3,259 | 3,256 | 3283 | 3,273 | 3,290 |
| Protein residues | 417 | 418 | 418 | 422 | 421 | 422 |
| Ligands | − | CHT: 1 | ETA: 1 | − | − | CHT: 1 |
| Average $B$ factors (Å²) | | | | | | |
| Protein | 87.45 | 40.28 | 50.80 | 76.93 | 82.55 | 63.63 |
| Ligand | − | 23.94 | 26.59 | − | − | 31.72 |
| R.m.s. deviations | | | | | | |
| Bond lengths (Å) | 0.004 | 0.004 | 0.002 | 0.003 | 0.004 | 0.003 |
| Bond angles (°) | 0.967 | 0.922 | 0.529 | 0.512 | 0.592 | 0.641 |
| Validation | | | | | | |
| MolProbity score | 1.50 | 1.40 | 1.40 | 1.20 | 1.31 | 1.31 |
| Clashscore | 6.84 | 7.25 | 5.14 | 4.20 | 5.71 | 5.68 |
| Poor rotamers (%) | 0.0 | 0.0 | 0.0 | 0.6 | 0.0 | 2.8 |
| Ramachandran plot | | | | | | |
| Favored (%) | 97.3 | 98.1 | 97.4 | 99.0 | 98.6 | 99.0 |
| Allowed (%) | 2.7 | 1.9 | 2.6 | 1.0 | 1.4 | 1.0 |
| Disallowed (%) | 0.0 | 0.0 | 0.0 | 0.0 | 0.0 | 0.0 |

This table provides comprehensive information on the Cryo-EM data collection, refinement and validation statistics for the studied samples. -, not applicable.

# Reporting Summary

## Statistics

For all statistical analyses, confirm that the following items are present in the figure legend, table legend, main text, or Methods section.

| n/a | Confirmed | |
|---|---|---|
| ☐ | ☒ | The exact sample size (*n*) for each experimental group/condition, given as a discrete number and unit of measurement |
| ☐ | ☒ | A statement on whether measurements were taken from distinct samples or whether the same sample was measured repeatedly |
| ☐ | ☒ | The statistical test(s) used AND whether they are one- or two-sided *Only common tests should be described solely by name; describe more complex techniques in the Methods section.* |
| ☒ | ☐ | A description of all covariates tested |
| ☐ | ☒ | A description of any assumptions or corrections, such as tests of normality and adjustment for multiple comparisons |
| ☐ | ☒ | A full description of the statistical parameters including central tendency (e.g. means) or other basic estimates (e.g. regression coefficient) AND variation (e.g. standard deviation) or associated estimates of uncertainty (e.g. confidence intervals) |
| ☐ | ☒ | For null hypothesis testing, the test statistic (e.g. *F*, *t*, *r*) with confidence intervals, effect sizes, degrees of freedom and *P* value noted *Give P values as exact values whenever suitable.* |
| ☐ | ☒ | For Bayesian analysis, information on the choice of priors and Markov chain Monte Carlo settings |
| ☒ | ☐ | For hierarchical and complex designs, identification of the appropriate level for tests and full reporting of outcomes |
| ☒ | ☐ | Estimates of effect sizes (e.g. Cohen's *d*, Pearson's *r*), indicating how they were calculated |

*Our web collection on statistics for biologists contains articles on many of the points above.*

## Software and code

Policy information about availability of computer code

| | |
|---|---|
| Data collection | The radioactivity assays were quantified by scintillation counter Tri-Carb (Perkin Elmer). The immunofluorescent staining samples were imaged with laser confocal microscopes (Zeiss LSM710 and Leica STELLARIS 5). Titan Krios microscopes operated at 300 kV and equipped with a Selectris X imaging filter and a Falcon4 camera (Thermo Fisher) or a BioQuantum energy filter and a K3 camera (Gatan); Data collection quality was monitored through EPU v. 3.0-3.4 and CryoSparc Live v3.0 and 4.0. Differential scanning fluorimetry of purified FLVCR variants were investigated with a Prometheus Panta (NanoTemper, no other version available). All molecular dynamics simulations were performed using the GROMACS v.2022.4 software. Biorender was used to draw cartoon and illustration. |
| Data analysis | For cryo-EM data analysis: ChimeraX v.1.5 and 1.6; MotionCor2-2.1.2.6; Gctf v.1.06; CLUSTAL Omega v.1.2.4; RELION-3.1 and -4.0; CryoSPARC Live v3.0 and v4.0; COOT v. 0.8.9; Phenix (v1.18); MolProbity v.4.5.  For molecular dynamics simulation analysis: CHARMM36m force field was used for protein, lipids, heme and ions, together with TIP3P water; LINCS algorithm was used to constrain the bonds involving hydrogen atoms; Visual Molecular Dynamics (VMD) v.1.9.3 and GROMACS v.2022.4; PROPKA server v.3.5.0 was used to predicts the pKa values of ionizable groups; Pymol v.3.0 was used to introduce mutations; gmx_MMPBSA v1.6.2 was used for energy calculation. NanoDSF data was analyzed via PR.Panta Analysis v1.4.4 and Python libraries numpy v.1.21.5, scipy v.1.7.3, pandas v.1.5.2, matplotlib v.3.6.2, and seaborn v.0.12.2 in Microsoft Visual Studio Code v.1.80.1. Tunnels and cavities were mapped with MOLE 2.5, and calculated with CASTp. ConSurf server (https://consurf.tau.ac.il/) was used for conservation analysis. Online version of BioRender was used for illustration generation. |

For manuscripts utilizing custom algorithms or software that are central to the research but not yet described in published literature, software must be made available to editors and reviewers. We strongly encourage code deposition in a community repository (e.g. GitHub). See the Nature Portfolio guidelines for submitting code & software for further information.

## Data

Policy information about availability of data

All manuscripts must include a data availability statement. This statement should provide the following information, where applicable:
- Accession codes, unique identifiers, or web links for publicly available datasets
- A description of any restrictions on data availability
- For clinical datasets or third party data, please ensure that the statement adheres to our policy

Cryo-EM maps are deposited to the Electron Microscopy Data Bank under accession numbers: EMD-18334, EMD-18335, EMD-18336, EMD-18337, EMD-18339, EMD-19009. Atomic models of human FLVCR1 and FLVCR2 have been deposited to the Protein Data Bank under accession numbers: 8QCS, 8QCT, 8QCX, 8QCY, 8QD0, 8R8T. All molecular dynamics trajectories generated for this study and simulation input files are deposited in a Zenodo repository and freely available via the following DOI: 10.5281/zenodo.10952971. All other data is presented in the main text or supplementary materials. Source data are provided with this paper.

## Research involving human participants, their data, or biological material

Policy information about studies with human participants or human data. See also policy information about sex, gender (identity/presentation), and sexual orientation and race, ethnicity and racism.

| | |
|---|---|
| Reporting on sex and gender | N/A |
| Reporting on race, ethnicity, or other socially relevant groupings | N/A |
| Population characteristics | N/A |
| Recruitment | N/A |
| Ethics oversight | N/A |

Note that full information on the approval of the study protocol must also be provided in the manuscript.

# Field-specific reporting

Please select the one below that is the best fit for your research. If you are not sure, read the appropriate sections before making your selection.

☒ Life sciences          ☐ Behavioural & social sciences          ☐ Ecological, evolutionary & environmental sciences

For a reference copy of the document with all sections, see nature.com/documents/nr-reporting-summary-flat.pdf

# Life sciences study design

All studies must disclose on these points even when the disclosure is negative.

| | |
|---|---|
| Sample size | Sample sizes (number of collected micrographs) of respective cryo-EM datasets were chosen based on instrument availability and experimental design. Datasets of > 5000 micrographs ensured a sufficient number of particles to achieve resolutions < 3.5 Å. The smallest collected dataset contained 5356 micrographs (FLVCR2 supplemented with heme dataset) while the largest dataset contained 14014 micrographs (FLVCR1 supplemented with choline dataset). Nano differential scanning fluorimetry were performed in technical replicates (NanoDSF, n = 3). Technical replicates were chosen to determine standard deviation values for each data points and to validate data quality. For functional assays, no predetermined sample size was applied. For these experiments, 3 biological replicates were always used in each experiment. Most of the functional assays were repeated 2-3 times. However, only one dataset was shown in the manuscript. |
| Data exclusions | No data were excluded. |
| Replication | Single particle cryo-EM is based on averaging protein particles of nearly identical orientation within a vitreous layer of ice. Therefore, replication is not per se required to ensure statistical robustness of structural data. In case of this work, we have determined 2 individual structures of FLVCR1 and 4 individual structures of FLVCR2 under different sample conditions. Hence these data can be considered as biological replicates of the presented structural data. |
| | For the functional assays (testing the activity of the mutants), we have performed the experiments with at least 3 biological replicates using different concentrations of choline and ethanolamine (the ligands). However, we chose to use the concentrations in which the datasets are used in the current manuscript. The results of transport assays using different concentrations of choline and ethanolamine are also consistent with the results used in the manuscript. |
| | For the other functional assays, we repeated the experiments using the same conditions at least twice.  These results are also included in the SOURCE data file, but please note that only one dataset was used to draw the graphs in the manuscript. |
| Randomization | Generally, no randomization was required for the experimental design of this study. However, it is to note that particles are randomized during data processing steps in Relion and cryoSPARC (randomization during 2D classification, randomized half sets during Refine3D). Randomized half sets of particles are used in final reconstruction steps in order to determined gold-standard Fourier shell correlations based |

on the 0.143 level.
For the functional assays, Western blot analysis and IF of the mutants, we did not randomize the samples as the ID of the mutant plasmids used for transfection are known by the experimentalists.

Blinding | For all studies in this manuscript, such as 3D reconstruction, or NanoDSF measurements, there was no awareness of group assignment that caused biased results, so blinding was was relevant for data reliability.

# Reporting for specific materials, systems and methods

We require information from authors about some types of materials, experimental systems and methods used in many studies. Here, indicate whether each material, system or method listed is relevant to your study. If you are not sure if a list item applies to your research, read the appropriate section before selecting a response.

## Materials & experimental systems

| n/a | Involved in the study |
|---|---|
| ☐ | ☒ Antibodies |
| ☐ | ☒ Eukaryotic cell lines |
| ☒ | ☐ Palaeontology and archaeology |
| ☐ | ☒ Animals and other organisms |
| ☒ | ☐ Clinical data |
| ☒ | ☐ Dual use research of concern |
| ☒ | ☐ Plants |

## Methods

| n/a | Involved in the study |
|---|---|
| ☒ | ☐ ChIP-seq |
| ☒ | ☐ Flow cytometry |
| ☒ | ☐ MRI-based neuroimaging |

## Antibodies

Antibodies used | Monoclonal ANTI-FLAG® M2 antibody produced in mouse - clone M2, purified immunoglobulin (Purified IgG1 subclass), buffered aqueous solution (10 mM sodium phosphate, 150 mM NaCl, pH 7.4, containing 0.02% sodium azide), Merck (formly Sigma-Aldrich), F3165; Anti-Mouse IgG (whole molecule) - Alkaline Phosphatase antibody produced in goat, Merck (formly Sigma-Aldrich), A9316. FLAG-tagged FLVCR1 and FLVCR2 were detected using anti-FLAG (F3165, Sigma-Aldrich) antibodies at 1:1,000 dilution. Anti-mouse IgG antibody conjugated with alkaline phosphatase (A9316, Sigma-Aldrich) was used as secondary antibody at 1:5,000 dilution. Native FLVCR1 and FLVCR2 proteins were detected by polyclonal FLVCR1 and FLVCR2 antibodies raised in-house at 1:1,000 dilution. GAPDH antibody (sc-32233, Santa Cruz) was used at 1:4,000 dilution. Monoclonal ANTI-FLAG® M2-FITC antibody produced in mouse (F4049, Sigma-Aldrich) was used at final concentration of 10 µg/mL in TBS for immunofluorescent staining against the FLAG-tags in the overproduction stable cells. Alexa Fluor 555 (A-21428, Invitrogen™, ThermoFisher, Lot 2527964, polyclonal Goat anti-Rabbit IgG) was used as secondary antibody at 1:500 dilutions for immunofluorescent staining against the polyclonal FLVCR1 and FLVCR2 primary antibodies for HEK293 cells.
Details of other antibodies used in the study was included in the manuscript as well.

Validation | Primary antibody Monoclonal ANTI-FLAG® M2 antibody produced in mouse was validated by the manufacturer:
Sensitivity Test - Detects 2 ng of FLAG-BAP fusion protein by dot blot using chemiluminescent detection.
Specificity - Detects a single band of protein on a western blot from an E. coli crude cell lysate.
For detailed information please see: https://www.sigmaaldrich.com/DE/en/product/sigma/f3165
Primary antibody Monoclonal ANTI-FLAG® M2-FITC antibody was validated by the manufacturer:
Immunoflourescence (Direct) performed using CMV-2 transfected COS-7 cells.
For detailed information please see: https://www.sigmaaldrich.com/DE/en/product/sigma/f4049
Primary antibody polyclonal FLVCR1 and FLVCR2 antibodies against human proteins were validated using cells with and without FLVCR1 and FLVCR2 genes.

## Eukaryotic cell lines

Policy information about cell lines and Sex and Gender in Research

Cell line source(s) | Flp-In™ T-REx™ 293 Cell Line: cells are purchased from Thermo Fisher (formly Invitrogen™). HEK293 cells originally from ATCC (CRL-1573) were used.

Authentication | No further authentication was performed for the commercially available cell line.

Mycoplasma contamination | Periodically test negative.

Commonly misidentified lines
(See ICLAC register) | No commonly misidentified cell lines were used in this study.

# Animals and other research organisms

| | |
|---|---|
| Laboratory animals | Mouse tissues were used in the study. These mice include FLVCR1f/f-Mx1Cre and control mice (FLVCR1f/+Mx1Cre and FLVCR1f/f). Tissues were collected from mice at 3-6 months old. |
| Wild animals | No wild animal was used in this study. |
| Reporting on sex | Gender of mice was not reported. |
| Field-collected samples | No field collected sample was used in this study. |
| Ethics oversight | Studies involving mice were reviewed and approved by the University of Washington Institutional Animal Care and Use Committee under protocol number 2001-13. |

Note that full information on the approval of the study protocol must also be provided in the manuscript.

