## [Peer Review File · Nature]

Manuscript Title: Molecular mechanism of choline and ethanolamine transport in humans

Reviewer Comments & Author Rebuttals

Reviewer Reports on the Initial Version:

Referees' comments:

Referee #1 (Remarks to the Author):

In this contribution by Ri et al., a comprehensive functional and structural analysis of two SLC49 family transporters is reported. The strength and rigour of this study originates from an integrative approach. The characterised transporters belong to MFS superfamily and (obviously not very surprisingly) FLVCR1 and 2 have been shown to operate in a rocker switch mechanism. However, one of the exciting findings is that these transporters are apparently involved into the transport of ethanolamine. Furthermore, the authors provide a compelling evidence that choline and not heme is the transported substrate.

The analysis of provided models and corresponding maps revealed the good agreement between experimental data and models with a few exceptions. For example, the modelling of a peripheral heme binding is not convincing, the density proposed for a bound heme does not allow unambiguous assignment of a heme and looks more like a lipid mixture. The mode of interaction between the propionate and Arg side chain is incapable to provide any strong interaction anyway (which is somehow also confirmed by MD). Hence this part of the manuscript should be either entirely removed or the authors should spend more time using other techniques (ITC, TSA, spectroscopy etc) to confirm these interactions. Furthermore, there is basically no density at the glycosylated sites in FLVCR1-IF-choline_8QCT.pdb and FLVCR1-IF-apo_8QCS.pdb.

Furthermore I have a question regarding EM data processing. I was a bit confused with the fact that at several instances the authors proceeded with the minor classes. For example in Suppl Fig2 for FLVCR1 processing panel a: the authors took the last class (14%) but ignoring other classes (including the most populated one with 55.9%); in panel b: they ignored the class with 85.6% and took the one with 4.6% and similar for panel c.

The same is in principle true for the rest of the processing. In my view the authors should carefully reevaluate their data and perform also 3Dflex refinement.

Minor concerns:

- In some figure legends panel descriptions are scrambled (I think for sure in Fig1 of the main text)
- 'mock' should be defined much earlier. I think now it is only defined in Fig.3 as an empty vector
- line 218: 'two proximal asparagine' should be glutamine
- fig1 panels c,d,f,g - #of data points is quite minimal
- Extended F2: 'Relative' - I guess relative?
- Fig.3 panel h: it is impossible to discern FLVCR1 from FLVCR2 data points - perhaps use a different colour.
- Fig.3: I failed to grasp the panels f and g and why distances vary so much? In c and d panels the

distances are below 4 Å and in f and g it is above 50 Å.

Referee #2 (Remarks to the Author):

This manuscript employs a combination of cell-based transport assays, structural biology, and molecular dynamics simulations to establish that FLVCR1 and FLVCR2, two transporters that were previously linked to heme transport, are, in fact, choline/ethanolamine transporters. Using cryo-EM, the authors elucidated the architectures of FLVCR1 in the inward-facing apo and substrate-bound states and FLVCR2 in the inward-facing apo, outward-facing apo, and inward-facing substrate-bound states.

The structural and functional characterization of the choline binding and transport by these transporters is well done, and the conclusion is well supported by the data. The mechanistic insights will be valuable. However, I have two serious concerns: 1) Heme binding to FLVCR2, 2) Ambiguity concerning ethanolamine binding pose especially in light of the lack of acknowledgment of the strong density in the apo state. These two issues must be addressed.

Major points:

1. In the FLVCR1-IF-apo map, there is a substantial globular density close to the binding site residues W125 and Y349. This density is stronger than that of ethanolamine. Also, there is a similar density in the FLVCR2-apo map. However, the authors did not comment on this density and simply omitted any reference to this density, which is not appropriate. The authors must know they should present all the data. The authors must acknowledge that there is a strong density in the apo state and discuss/analyze what it is and how it does not affect the conclusion regarding the ligand binding poses that the authors propose.
 2. Ethanolamine is small and thus discerning its binding pose via cryo-EM is not a trivial task. In the FLVCR1-IF-ethanolamine cryo-EM map, the cryo-EM density for ethanolamine is weaker than that of apo state. This reviewer noticed that there are multiple ways for ethanolamine to be oriented but the authors did not test all the possible binding poses. In the FLVCR1-IF-ethanolamine model, the ethanolamine is built so that its hydroxyl group is facing Q214, and the amine group is closer to the W125. The amine group to the W125 is slightly different from that of choline and the authors attributed this difference in the binding pose as the difference in the selectivity in ligand recognition. However, in this case the distance between the amine and the TRP ring is too close (3.2 Å) for a cation- π interaction. It is also possible that the orientation of ethanolamine is inverted so that amine and hydroxy groups are changed. Also, depending on the angles of amine and hydroxy groups relative to the CH₂-CH₂ group, they can form H-bond to either Q214 or Q471. Have the authors also tried inverting amine and the hydroxyl group of the molecule? Have they also tried to change the rotation of the amine and hydroxy groups? Authors should test MD simulations and if possible, try to calculate free energy to derive the best binding pose for ethanolamine.
- Finally, regardless of the support from the MD result, the authors need to tone down that the accurate pose for ethanolamine is difficult to probe.
3. The cryo-EM density of heme in the FLVCR2-OF-heme map is, at the very least, ambiguous. The signal for the central iron is absent and the iron lacks additional coordination, which is inconsistent with the typical heme density. Based on the current model, only R83 may have some interaction. R82 and K273

are both too far from the heme to have electrostatic interaction, as described in the manuscript. Furthermore, in this pose, the hydrophobic part of the porphyrin ring is exposed to the solvent, making it energetically unfavorable. The EM density in the FLVCR2-IF-heme-choline map is even worse and it is not distinguishable from nearby noise. In supplementary figure 10, the stable pose of heme from the MD simulation is not convincing either. In this pose, heme acts as if a detergent, with the polar part of the porphyrin ring facing the solvent and the hydrophobic part buried in the micelle. However, the central iron still lacks coordination and the positively charged iron is unlikely to be stable in the micelle. My opinion is that all proposed binding poses are unrealistic and are unlikely to make any significant impact on the transporter.

The author's observation of the cryo-EM population shift in the presence of heme is not consistent with the functional data. It is not uncommon that prep-to-prep variation or some minor changes in the experimental condition to affect the cryo-EM population shift or result in differences in 3D classification. As such, it cannot be direct evidence for its binding to the transporter, also the sample size is limited (n=1).

The current interpretation of the cryo-EM density as heme is not acceptable to anyone's standard. If the authors want to maintain the argument that the spurious density is indeed heme, the authors should perform direct binding experiments coupled with mutagenesis (e.g. ITC). Also, I believe that the authors must deposit the coordinates with heme so that the colleague scientists in the field can assess the data without bias. If not, I strongly suggest that the authors remove the interpretation of the heme binding site and associated cryo-EM data and rewrite the manuscript. In my opinion, removing the unsubstantiated heme data and retaining the functional data would not change the conclusion of the manuscript. In fact, I think it will strengthen the overall quality of the manuscript.

Minor comments

1. Could the authors specify which data corresponds to the statement in Line 113 "aided by the typically negative membrane potential"? This part is not explained clearly in the manuscript.
2. In Line 180, "Our mutagenesis studies underscore a greater significance of S203 compared to R200". First, why does the left panel of Extended Data Fig 5b have two sets of mock and FLVCR2 WT data with one set having multiple mutants and the other set having only S203Y? In these two sets of data, the WT seems to have different fold of change. If this is due to two separate experiments, the authors need to explicit it in the figure legend. Second, R200A was mutated to alanine and S203 to tyrosine. Admittedly the effect of the mutants on choline transport is different. However, the effect is likely dependent on what amino acid the residues are mutated to. The reviewer thinks that it is not justifiable to say one mutant has greater significance than the other unless the authors compare the effect of multiple mutations of these two residues.
3. Line 218, "two proximal asparagine". Based on the text, it should be glutamine.
4. In the title of Extended Data Fig.8. Q214F is a typo.
3. In the method session, line 395. Please specify what method was used to determine total protein levels.
4. In Extended Data Fig 2, please specify what GPC and GPE are.
5. Line 305, F348F is a typo.

Referee #3 (Remarks to the Author):

Work by Ri and colleagues focus on FLVCR proteins. It has been suggested earlier that FLVCR1/2 are heme transporters, but they may also catalyze choline and ethanolamine transport. The evidence for choline transport by FLVCR1 has been obtained recently. Here authors have used several experimental and computational techniques to characterize the mechanism of FLVCR transporters. This is a very interesting work and can be published, but I would like to see a more involved computational analysis. My major and minor comments are below.

Major points

It has been shown in this work that pH gradients do not play a role in transport mechanisms of FLVCRs. But it is the downhill gradient that is responsible for transport. However, protonation states of amino acid residues are important for protein function. Did authors perform basic pKa analysis prior to MD simulations? If not, this would be highly recommended. For many proteins, a large effect is seen if MD simulations are performed with all titratable amino acids modeled in charged states vs charged states modified after pKa calculations. See here for instance - <https://doi.org/10.1101/2023.08.24.554589>. If authors have not performed pKa analysis, maybe some MD simulations after pKa analysis would be important to perform and show that this does not change their conclusions. It is understandable that there may not be many titratable residues that undergo pKa change.

In very nice extended data fig. 5 (and others), several residues are shown and it is clear how distance between several residues change in different conformations. It is also very interesting to note (often not emphasized enough) a strong interaction between Ser-Glu residues (S203/E435) is stabilized in structure. Authors can plot the distribution of the distances between these different residues and show how they differ between different states. This will consolidate their findings. Furthermore, if backbones/sidechains remain in a conformation seen in structure (during MD simulations), this will synergistically consolidate the structural data and MD simulations (see also - <https://doi.org/10.1101/2023.08.24.554589>)

Authors have mentioned in the methods section that for almost all MD simulations three independent replicas have been initiated and that are 1 us long. This is good statistics. However, figures like 3c, 3d, 4b, 4d, including some in extended data/SI do not show all replicas. It would be important to show this

data that all replicas behave similarly. This will strengthen the conclusions even more.

Authors can get more quantitative estimates from simulations - I do not think any free energy calculations are needed at this stage (which can be a future work, as this is predominantly a solid experimental work supported by MD simulations), but for instance on the selectivity of choline or ethanolamine in the pockets(?) Authors can perform some basic interaction energy calculations on simulation snapshots and perhaps observe some differences. MM-GBSA/PBSA-like calculations (if possible on these type of systems) can also be tested. Furthermore, they can evaluate how much a mutation destabilizes the binding of ligand (based on force field based interaction energies, e.g.). Analysis of such type will have a far reaching impact on their simulation-based findings and set the stage for future explorations on these systems by more advanced simulation methods.

Cation- π interactions are observed to be stabilized in MD simulations. In Fig. 3c,d how is the distance measured? Is this the center of mass of the ring of Tyr/Trp and N atom of choline? Similarly, what is the distance measured in 4b?

Minor points

Were there any missing parts in the PDB file used for MD simulations? Were they modeled? This should be clarified if that is the case. For instance, some parts with weak density are present in the map (lines 126-127).

Was CHARMM force field used for choline and ethanolamine?

Was glycosylation modeled in the MD simulations? If not, do authors expect its modeling may affect their conclusions?

The lipid bilayer composition used is physiologically relevant(?) It is well-known that lipid bilayer composition can affect the conformational dynamics of proteins (e.g. GPCRs).

Author Rebuttals to Initial Comments:

Referees' comments:

Referee #1 (Remarks to the Author):

In this contribution by Ri et al., a comprehensive functional and structural analysis of two SLC49 family transporters is reported. The strength and rigour of this study originates from an integrative approach. The characterised transporters belong to MFS superfamily and (obviously not very surprisingly) FLVCR1 and 2 have been shown to operate in a rocker switch mechanism. However one of the exciting findings is that these transporters are apparently involved into the transport of ethanolamine. Furthermore, the authors provide a compelling evidence that choline and not heme is the transported substrate.

The analysis of provided models and corresponding maps revealed the good agreement between experimental data and models with a few exceptions. For example, the modelling of a peripheral heme binding is not convincing, the density proposed for a bound heme does not allow unambiguous assignment of a heme and looks more like a lipid mixture. The mode of interaction between the propionate and Arg side chain is incapable to provide any strong interaction anyway (which is somehow also confirmed by MD). Hence this part of the manuscript should be either entirely removed or the authors should spend more time using other techniques (ITC, TSA, spectroscopy etc) to confirm these interactions.

- We thank the reviewer for the constructive criticisms and highly value the comments on our data. Following the suggestion of our reviewers, and from our editor, we have removed the part on the heme binding-site. In this way, our manuscript becomes more focused and centers more around the core biological context of choline and ethanolamine transport.

Furthermore, there is basically no density at the glycosylated sites in FLVCR1-IF-choline_8QCT.pdb and FLVCR1-IF-apo_8QCS.pdb.

- We appreciate the reviewer's insightful feedback on our model construction. Although the glycan density features are not resolved with utmost precision, we here highlight the distinct density characteristics surrounding the asparagine residue N265, compared to the nearby asparagine group N268 (Rev. fig. 1). Nonetheless, after thorough deliberation, we have opted to exclude the glycan group from our PDB model submission to prevent potential bias.

- Additionally, we have made the decision to omit the section on FLVCR glycosylation from the manuscript, as it does not directly advance the molecular-level understanding of these transporters in our current study. This aspect of FLVCRs will be revisited in future research.

Furthermore, I have a question regarding EM data processing. I was a bit confused with the fact that at several instances the authors proceeded with the minor classes. For example in Suppl Fig2 for FLVCR1 processing panel a: the authors took the last class (14%) but ignoring other classes (including the most populated one with 55.9%); in panel b: they ignored the class with 85.6% and took the one with 4.6% and similar for panel c. The same is in principle true for the rest of the processing. In my view the authors should carefully reevaluate their data and perform also 3Dflex refinement.

- We thank the reviewer for the very careful and considerate evaluation of our cryo-EM data processing. In response to the inquiry about our classification subset selection criteria, we wish to clarify that following each 3D classification round—regardless of alignment or simple sorting—we carefully assessed the resulting density maps for quality and resolution through both metrics and visual inspection, advancing only those maps that seemed promising for further refinement stages.
- In cases where we selected low-abundance particle classes, the resulting maps showed marked improvement in quality compared to those derived from more abundant classes, following our selection criteria. It's important to note that for each of these instances, we had already conducted refinement runs using the entire particle stack before proceeding with 3D sorting, which led to notable resolution improvements when selecting these low-abundance, high-quality classes. We realize that we had originally omitted this step from our processing pipeline figure in favor of simplicity. We have now updated S. Figs. 2 & 3 and show the results of the full stack refinements prior to sorting steps.
- Following the reviewer's request, we revisited our methodology, conducting refinement for all individual sub-classified particle stacks. In conclusion, our in-depth re-evaluation confirmed that our initial selection criteria yielded the highest-quality maps (Rev. fig. 2).

Revision figure 2 – Re-evaluation of data processing steps

- In addition, we have also performed 3D flex refinement with the full particle stacks following the suggestion of the reviewer. In summary, flexibility refinement did neither result in higher resolution maps for the individual datasets, nor did we observe prominent flexibility that would advance our understanding regarding the dynamic nature of FLVCRs (Rev. fig. 3). Hence, we decided not to include these results in our manuscript.

Revision figure 3 – 3DFlex refinement for all datasets with the particle stacks before 3D sorting

Minor concerns:

- In some figure legends panel descriptions are scrambled (I think for sure in Fig1 of the main text)

- We thank reviewer for pointing this out and have corrected the figure legends panel description in Fig. 1 accordingly. We have also further reviewed all other figure legends to avoid and correct any issues.

- 'mock' should be defined much earlier. I think now it is only defined in Fig.3 as an empty vector

- We have updated the figure 1 caption to define mock. The caption now includes the sentence: "The inactive S203Y mutant of FLVCR2, and empty vector (mock) are used as controls."

-line 218: 'two proximal asparagine' should be glutamine

- We thank the reviewer and have corrected this to glutamine. The sentence now reads: "In our choline simulations, this hydroxyl shows versatile interactions by forming transient hydrogen bonds with two proximal glutamine residues (Q214^{FLVCR1}/Q471^{FLVCR1}, Q191^{FLVCR2}/Q447^{FLVCR2})."

-fig1 panels c,d,f,g - #of data points is quite minimal

- We thank the reviewer for the comment. For these figures, we have performed the experiments using 4 data points (e.g. 4 times and 4 doses). We believe that these data points are sufficient for us to calculate the K_m and V_{max} of the transporters under this condition. We believe that adding 1-2 data points to the curves would not add much information.

If the reviewer was concerned about the replicates in these panels, we have performed the experiments twice with 3 biological replicates in each batch. However, we used the data from 1 dataset to draw the graphs in these panels. In these graphs, there are 3 different biological replicates per data point. We would not like to combine the data from these two datasets to generate the graphs because the values were slightly different from batch to batch. Nevertheless, these results are fully reproducible. We would like to insert the results from the second dataset below for your perusal (Rev. fig. 4).

Revision figure 4 – The second dataset of dose curves and time courses for choline and ethanolamine transport activities

- Extended F2: 'Relative' - I guess relative?

- This typo is now corrected.

-Fig.3 panel h: it is impossible to discern FLVCR1 from FLVCR2 data points - perhaps use a different colour.

- We would prefer to keep the current color code as it matches the color we have chosen for presentation of all FLVCR1 and FLVCR2 related functional and structural data throughout the manuscript. Instead, we have increased the line width of the symbols and trend lines in panel h for an improved clarity.

- Fig.3: I failed to grasp the panels f and g and why distances vary so much? In c and d panels the distances are below 4 Å and in f and g it is above 50 Å.

- We thank the reviewer for bringing up this point. While panels c/d show the simulated distances of bound choline to the tryptophan and tyrosine residues in the binding site of wild type transporters, panels f and g show the equivalent distance plots for the W125A (FLVCR1) and W102A (FLVCR2) mutants, respectively. For clarity we have updated the figure caption to clarify the difference between wild type and mutant variant simulations. The caption now reads:

“Minimum atom-pair distances between choline and the highly conserved tryptophan and tyrosine side chains forming the choline-binding pockets of **wild type** FLVCR1 (c) and FLVCR2 (d) in 1 μ s MD simulation runs. A cation- π interaction is assumed for distance < 4 Å (grey dashed line).” and “Distance plot of choline within the binding site of W125A^{FLVCR1} (f) and W102A^{FLVCR2} (g) **mutant variants** as function of time in MD simulations (left), and choline occupancy in binding site for WT and alanine mutants (right).”

Referee #2 (Remarks to the Author):

This manuscript employs a combination of cell-based transport assays, structural biology, and molecular dynamics simulations to establish that FLVCR1 and FLVCR2, two transporters that were previously linked to heme transport, are, in fact, choline/ethanolamine transporters. Using cryo-EM, the authors

elucidated the architectures of FLVCR1 in the inward-facing apo and substrate-bound states and FLVCR2 in the inward-facing apo, outward-facing apo, and inward-facing substrate-bound states.

The structural and functional characterization of the choline binding and transport by these transporters is well done, and the conclusion is well supported by the data. The mechanistic insights will be valuable. However, I have two serious concerns: 1) Heme binding to FLVCR2, 2) Ambiguity concerning ethanolamine binding pose especially in light of the lack of acknowledgment of the strong density in the apo state. These two issues must be addressed.

Major points:

1. In the FLVCR1-IF-apo map, there is a substantial globular density close to the binding site residues W125 and Y349. This density is stronger than that of ethanolamine. Also, there is a similar density in the FLVCR2-apo map. However, the authors did not comment on this density and simply omitted any reference to this density, which is not appropriate. The authors must know they should present all the data. The authors must acknowledge that there is a strong density in the apo state and discuss/analyze what it is and how it does not affect the conclusion regarding the ligand binding poses that the authors propose.

- We thank the reviewer for the comment regarding the extra density. We re-evaluated our cryo-EM density maps and came to the conclusion that only the *as isolated* FLVCR1 shows a discernable density within the binding pocket. In the case of FLVCR2 we observe density in this region only at noise level, suggesting that even if there might be a residual molecule bound to this site, it has low abundance. Hence, we decided to focus only on the additional density within the structure of *as isolated* FLVCR1. To avoid any potential bias, we have decided to refer to our structures determined in the absence of exogenous choline and ethanolamine with the term “as isolated” in all texts and figures. Within the manuscript text, we then also clarify which of these maps we identify as apo states, and which ones contain an extra non-protein density within the binding pocket (see comment further below).
- Furthermore, for the *as isolated* FLVCR1 cryo-EM density map, we have conducted an additional round of masked 3D sorting and were able to obtain a class with noticeably reduced density (Rev. fig. 5).

Revision figure 5 – Masked 3D sorting focusing on the ligand-binding site in FLVCR-IF^{as isolated} cryo-EM data

- However, we still decided to be very transparent about the density found within the binding cleft of the *as isolated* FLVCR1 map, as it is also a distinguishing feature between the structures of FLVCR1 and FLVCR2. When comparing the position of this extra density in the *as isolated* FLVCR1 map and the choline/ethanolamine FLVCR1 maps, it became apparent that this density might represent co-purified transport substrate. Whether the density represents a blend of choline and ethanolamine, or an additional unknown substrate molecule, remains elusive at this time. For clarity, we have prepared a figure highlighting the positions, sizes and shapes of

choline, ethanolamine and the unknown density in our FLVCR1 structures at identical resolution cutoffs and comparable map contour levels. We have included this figure to the supplementary information file of our manuscript (Rev. fig. 6).

Revision figure 6 – Cryo-EM density of the ligand-binding site in FLVCR1 and FLVCR2 (see also Supplementary Fig. 8)

- Further, as suggested by the reviewer, we have added a paragraph to our manuscript describing this observation in full transparency. It reads as follows:

“Of note, while the *as isolated* inward-facing and outward-facing conformations of FLVCR2 represent apo states of the transporter, we observed an extra non-protein density within the substrate binding cleft of the *as isolated* FLVCR1 structure. Introducing choline and ethanolamine to FLVCR1 leads to displacement of this unidentified molecule, evident from the altered size and shape of the specific densities for choline and ethanolamine in the respective EM maps. The precise nature of this ligand remains unknown. It is however conceivable that FLVCR1 is co-purified with a blend of choline and ethanolamine, creating an indistinct density for these ligands, or alternatively with a yet unidentified ligand.

2. Ethanolamine is small and thus discerning its binding pose via cryo-EM is not a trivial task. In the FLVCR1-IF-ethanolamine cryo-EM map, the cryo-EM density for ethanolamine is weaker than that of apo state. This reviewer noticed that here are multiple ways for ethanolamine can be oriented but the authors did not test all the possible binding poses. In the FLVCR1-IF-ethanolamine model, the ethanolamine is built so that its hydroxyl group is facing Q214, and the amine group is closer to the W125. The amine group to the W125 is slightly different from that of choline and the authors attributed this difference in the binding pose as the difference in the selectivity in ligand recognition. However, in this case the distance between the amine and the TRP ring is too close (3.2 Å) for a cation-pi interaction. It is also possible that the orientation of ethanolamine is inverted so that amine and hydroxy groups are changed. Also, depending on the angles of amine and hydroxy groups relative to the CH₂-CH₂ group, they can form H-bond to either Q214 or Q471. Have the authors also tried inverting amine and the hydroxyl group of the molecule? Have they also tried change the rotation of the amine and hydroxy groups Authors should test MD simulations and if possible, tries to calculate free energy to derive the best binding pose for ethanolamine.

- We thank the reviewer for the in-depth explanation of this point. To clarify these open questions, we have performed the following experiments and analyses and have come to a more refined conclusion:

Firstly, we have collected yet another dataset of the FLVCR1 sample in the presence of ethanolamine (ETA). Thus, we were able to arrive at an overall resolution of 2.9 Å with much clearer ETA density features. Accordingly, we have updated the ethanolamine model placement and respective figures in our manuscript (Rev. fig. 7).

Revision figure 7 – Cryo-EM density of the ligand-binding site from the previous and current merged dataset

Secondly, we have performed more in-depth analyses of our MD simulation runs to gain a deeper understanding of the binding dynamics of ethanolamine. These analyses led us to the conclusion that, in contrast to choline, ethanolamine binding is dynamic with the molecule adopting different poses within the binding pocket, with two distinct binding states (~98% and 2%) differing in the ethanolamine orientation, as shown in our updated Fig. 4 and Extended Data Fig. 9. In both states, the hydroxyl group of ethanolamine interacted with Q214^{FLVCR1} (Fig. 4c) and the primary amine maintained cation- π contacts with the conserved aromatics W125^{FLVCR1} and Y349^{FLVCR1} (Fig. 4d and Extended Data Fig. 9b,c). We note that the amide group at the tip of the long Q214^{FLVCR1} sidechain also moves and flips (Fig. 4b). In line with the dynamic nature of ethanolamine binding, the distance of the nitrogen atom of ethanolamine to W125 and Y349 is broadly distributed (Extended Data Fig. 9c and Rev. fig. 8).

Revision figure 8 – Distributions of distances between the N-atom of choline or ethanolamine and highly-conserved tryptophan and tyrosine residues.

The rapid sampling of distinct but consistent binding poses of ethanolamine in the simulations, and the population distribution in the binding pocket shown in Extended Data Fig. 9a, point to state 1 as the best binding pose for ethanolamine for our simulation model in terms of free energy.

Finally, regardless of the support from the MD result, the authors need to tone down that the accurate pose for ethanolamine is difficult to probe.

- We have followed the suggestion of our reviewer and present a revised paragraph with arguments that are more openly phrased, accounting for the possibility of more than one distinct binding mode, which is supported by our MD simulation results. The new paragraph now reads:

“The ethanolamine density was identified within the abovementioned ligand binding-pocket, occupying the same cleft site as choline. To characterize the coordination and binding dynamics of this structurally simpler substrate in more detail, we carried out MD simulations with ethanolamine in the binding site. In our simulations, ethanolamine exhibited pronounced dynamics, adopting different poses with two distinct binding state orientations. In both states, the hydroxyl group of ethanolamine interacted with Q214^{FLVCR1} and the primary amine group of ethanolamine largely maintained cation- π contacts with the conserved aromatics W125^{FLVCR1} and Y349^{FLVCR1}.”

3. The cryo-EM density of heme in the FLVCR2-OF-heme map is, at the very least, ambiguous. The signal for the central iron is absent and the iron lacks additional coordination, which is inconsistent with the typical heme density. Based on the current model, only R83 may have some interaction. R82 and K273 are both too far from the heme to have electrostatic interaction, as described in the manuscript. Furthermore, in this pose, the hydrophobic part of the porphyrin ring is exposed to the solvent, making it energetically unfavorable. The EM density in the FLVCR2-IF-heme-choline map is even worse and it is not distinguishable from nearby noise. In supplementary figure 10, the stable pose of heme from the MD simulation is not convincing either. In this pose, heme acts as if a detergent, with the polar part of the porphyrin ring facing the solvent and the hydrophobic part buried in the micelle. However, the central iron still lacks coordination and the positively charged iron is unlikely to be stable in the micelle. My opinion is that all proposed binding poses are unrealistic and are unlikely to make any significant impact on the transporter.

The author’s observation of the cryo-EM population shift in the presence of heme is not consistent with the functional data. It is not uncommon that prep-to-prep variation or some minor changes in the experimental condition to affect the cryo-EM population shift or result in differences in 3D classification. As such, it cannot be direct evidence for its binding to the transporter, also the sample size is limited (n=1).

The current interpretation of the cryo-EM density as heme is not acceptable to anyone’s standard. If the authors want to maintain the argument that the spurious density is indeed heme, the authors should perform direct binding experiments coupled with mutagenesis (e.g. ITC). Also, I believe that the authors must deposit the coordinates with heme so that the colleague scientists in the field can assess the data without bias. If not, I strongly suggest that the authors remove the interpretation of the heme binding site and associated cryo-EM data and rewrite the manuscript. In my opinion, removing the unsubstantiated heme data and retaining the functional data would not change the conclusion of the manuscript. In fact, I think it will strengthen the overall quality of the manuscript.

- We thank the reviewer for the constructive criticisms and highly value the comments on our data. Following the suggestion of our reviewers, and from our editor, we have decided to remove the part about the heme binding-site. In this way, our manuscript becomes more focused and centers more around the core biological context of choline and ethanolamine transport.

Minor comments

1. Could the authors specify which data corresponds to the statement in Line 113 “aided by the typically negative membrane potential”? This part is not explained clearly in the manuscript.

- We appreciate the reviewer's attention to detail and apologize for any confusion caused by the lack of specific data related to this statement. We wanted to emphasize that the transport mechanism of choline, being a positively charged ligand, follows a canonical potential-aided transport, as generally characterized in this field. However, we recognize that our data do not properly address this part, and have therefore decided to remove this statement from the manuscript. This aspect of FLVCRs will be revisited in future research.

2. In Line 180, “Our mutagenesis studies underscore a greater significance of S203 compared to R200”. First, why does the left panel of Extended Data Fig 5b have two sets of mock and FLVCR2 WT data with one set having multiple mutants and the other set having only S203Y? In these two sets of data, the WT seems to have different fold of change. If this is due to two separate experiments, the authors need to explicit it in the figure legend. Second, R200A was mutated to alanine and S203 to tyrosine. Admittedly the effect of the mutants on choline transport is different. However, the effect is likely dependent on what amino acid the residues are mutated to. The reviewer thinks that it is not justifiable to say one mutant has greater significance than the other unless the authors compare the effect of multiple mutations of these two residues.

- We thank the reviewer for the comment. We would like to clarify that there were two separate experiments performed for Extended Data Fig. 5b. Thus, there were two sets of mock and WT FLVCR2 controls that were used in these two panels, as stated in the original figure legend. We now explain this in the caption: “Separate mock experiments were performed in (b) and (c).”

With regard to the comparison of R200A and S203Y mutant, based on the fold change of these mutants to their respective mock, the R200A mutant still exhibits approximately 55% transport activity while S203Y showed complete loss of transport activity. This led us to draw the conclusion.

We agree with the reviewer that the transport activity of these mutants might be dependent on the substituted residue. Thus, we performed follow-up experiments to generate an S203A mutant and compared the transport activity of this mutant to R200A mutant. Our results showed that S203A also exhibited abolished transport activity similar to S203Y (Rev. fig. 9). These results show a critical involvement of S203 in choline transport activity in FLVCR2. We have revised the text accordingly. Please see below.

Revision figure 9 – Transport assay of corresponding FLVCR2 mutants (see also Extended Data Fig. 5)

3. Line 218, “two proximal asparagine”. Based on the text, it should be glutamine.

- We have corrected this to glutamine. The sentence now reads: “In our choline simulations, this hydroxyl shows versatile interactions by forming transient hydrogen bonds with two proximal glutamine residues (Q214^{FLVCR1}/Q471^{FLVCR1}, Q191^{FLVCR2}/Q447^{FLVCR2}).”

4. In the title of Extended Data Fig.8. Q214F is a typo.

- We have corrected this typo. The “F” should have been in superscript.

3. In the method session, line 395. Please specify what method was used to determine total protein levels.

- We apologize for the lack of the information. The method used for total protein assay was added to the method section. We have revised the text accordingly.

4. In Extended Data Fig 2, please specify what GPC and GPE are.

- We have revised the text accordingly.

5. Line 305, F348F is a typo.

- We have corrected this typo. The “F” should have been in superscript.

Referee #3 (Remarks to the Author):

Work by Ri and colleagues focus on FLVCR proteins. It has been suggested earlier that FLVCR1/2 are heme transporters, but they may also catalyze choline and ethanolamine transport. The evidence for choline transport by FLVCR1 has been obtained recently. Here authors have used several experimental and computational techniques to characterize the mechanism of FLVCR transporters. This is a very interesting work and can be published, but I would like to see a more involved computational analysis. My major and minor comments are below.

Major points

It has been shown in this work that pH gradients do not play a role in transport mechanisms of FLVCRs. But it is the downhill gradient that is responsible for transport. However, protonation states of amino acid residues are important for protein function. Did authors perform basic pKa analysis prior to MD simulations? If not, this would be highly recommended. For many proteins, a large effect is seen if MD simulations are performed with all titratable amino acids modeled in charged states vs charged states modified after pKa calculations. See here for instance - <https://doi.org/10.1101/2023.08.24.554589>. If authors have not performed pKa analysis, maybe some MD simulations after pKa analysis would be important to perform and show that this does not change their conclusions. It is understandable that there may not be many titratable residues that undergo pKa change.

- We thank the reviewer for emphasizing the importance of the protonation states of amino acid residues in protein function and for pointing out the necessity of pKa analysis in our methodological approach. We would like to clarify that pKa analysis was indeed conducted prior to the molecular dynamics (MD) simulations. This was inadvertently omitted from the methods section of our manuscript. We have since rectified this oversight and included a description of the pKa analysis procedure. It reads in the method section: “We set the protonation state of each residue as predicted for pH 7.0 using the PROPKA server.” We believe that this clarification will provide the reader with better understanding of our methodology and reinforce the robustness of our conclusions.

In very nice extended data fig. 5 (and others), several residues are shown and it is clear how distance between several residues change in different conformations. It is also very interesting to note (often not emphasized enough) a strong interaction between Ser-Glu residues (S203/E435) is stabilized in structure. Authors can plot the distribution of the distances between these different residues and show how they differ between different states. This will consolidate their findings. Furthermore, if backbones/sidechains remain in a conformation seen in structure (during MD simulations), this will synergistically consolidate the structural data and MD simulations (see also - <https://doi.org/10.1101/2023.08.24.554589>)

- We are grateful for the reviewer’s insightful comments and suggestions regarding the interactions between residues in different conformational states. In accordance with the reviewer’s recommendations, we have now included a detailed analysis of the distance distributions for key residue interactions. These are illustrated in the updated Supplementary Fig. 5c, which showcases the distribution of distances between the relevant residues across the different states observed in our study. This addition aims to provide a clearer and more quantitative demonstration of the conformational changes and interactions that the reviewer rightly highlighted as important for understanding the mechanism at play.

Authors have mentioned in the methods section that for almost all MD simulations three independent replicas have been initiated and that are 1 μ s long. This is good statistics. However, figures like 3c, 3d, 4b, 4d, including some in extended data/SI do not show all replicas. It would be important to show this data that all replicas behave similarly. This will strengthen the conclusions even more.

- We appreciate the reviewer's attention to detail regarding the representation of our simulation replicas and their significance to our conclusions. In response to this valuable suggestion, we have created a new supplementary figure to include the previously missing data from all replicas. This additional figure, which we have inserted as Supplementary Fig. 10, provides all the replicas that were not shown in the previous figures.
- Notably, this new figure also demonstrates the event where choline exits the cavity of FLVCR2 after 800 ns in replica 1, an event that is crucial to the release mechanism discussed in our manuscript. This behavior is clearly captured in the corresponding distance plot. Furthermore, we observed that ethanolamine demonstrates pronounced dynamics within the FLVCR1 cavity; particularly, in replica 3, it exits early in the simulation. While this variability suggests that the distance plot for this replica is less informative, it underscores the importance of examining multiple replicas to capture the full range of molecular behavior.

Authors can get more quantitative estimates from simulations - I do not think any free energy calculations are needed at this stage (which can be a future work, as this is predominantly a solid experimental work supported by MD simulations), but for instance on the selectivity of choline or ethanolamine in the pockets(?) Authors can perform some basic interaction energy calculations on simulation snapshots and perhaps observe some differences. MM-GBSA/PBSA-like calculations (if possible on these type of systems) can also be tested. Furthermore, they can evaluate how much a mutation destabilizes the binding of ligand (based on force field based interaction energies, e.g.). Analysis of such type will have a far reaching impact on their simulation-based findings and set the stage for future explorations on these systems by more advanced simulation methods.

- We thank the reviewer's valuable suggestion on free energy calculations. Guided by these suggestions, we performed MMPBSA on FLVCR1 and FLVCR2 with choline. We used a gromacs implementation of a very widely used AMBER protocol (gmx_mmpbsa). We used "interaction entropy" to estimate the entropic term (<https://pubs.acs.org/doi/full/10.1021/jacs.6b02682>).
- We used a dielectric constant of 7.0 for the membrane, 80.0 for the solvent and 4.0 for the protein. We also did alanine scanning for the conserved tryptophan residues and energy decomposition for the residues in the binding site, and in general the results qualitatively agree with the experimental observations.
- We included the results for choline and ethanolamine in Extended Data Fig. 8. The results for ethanolamine are not as clear as those for choline, probably due to the worse statistics resulting from the more dynamic binding mode of ethanolamine. Nevertheless, the influence of W125^{FLVCR1} and Q214^{FLVCR1} residue is still visible in these calculations in agreement with experimental results.

Cation-pi interactions are observed to be stabilized in MD simulations. In Fig. 3c,d how is the distance measured? Is this the center of mass of the ring of Tyr/Trp and N atom of choline? Similarly, what is the distance measured in 4b?

- We apologize for the lack of detailed information regarding the distance measurement methodology. We have included the relevant information in the respective figure captions. In Fig. 3c,d it reads now: "Minimum atom-pair distances between choline and the highly conserved tryptophan and tyrosine side chains forming the choline-binding pockets of wild type FLVCR1 (c) and FLVCR2 (d) in 1 μ s MD simulations." In Methods, we define "Minimum atom-pair distances were calculated as the minimum distance over all pairs of atoms in the two stated groups (e.g., ligand and certain defined sidechains)." For Fig. 4b, we have updated the caption: "Snapshots of the two distinct states of ethanolamine (left) and minimum atom-pair distance of

ethanolamine to conserved aromatic side chains of FLVCR1 as function of time in MD simulation (right).”

Minor points

Were there any missing parts in the PDB file used for MD simulations? Were they modeled? This should be clarified if that is the case. For instance, some parts with weak density are present in the map (lines 126-127).

- We appreciate the reviewer's request for clarification regarding the completeness of the protein structure used in our molecular dynamics (MD) simulations. In our runs, the terminal regions of the protein, which were not visible in the density map, were not modeled due to the lack of structural information. However, the rest of the protein structure, including regions with weak electron density, were carefully modeled to maintain the integrity of the functional domains and the overall protein conformation. The complete simulation setup will be deposited in a zenodo repository made openly accessible upon publication.

Was CHARMM force field used for choline and ethanolamine?

- We did apply the CHARMM36m force field in the MD simulations, and we have updated the method section to clarify this point: “The CHARMM36m force field was used with the improved WYF parameters for cation- π interactions, in particular of the choline and ethanolamine ligands.”

Was glycosylation modeled in the MD simulations? If not, do authors expect its modeling may affect their conclusions?

- We thank the reviewer for the question regarding the modeling of a possible glycosylation in our MD simulations. In our study, glycosylation was not explicitly modeled and the discussion of a possible glycosylation has been removed from the revised paper. In response to the reviewer question, we anticipate that the absence of this post-translational modification does not impact our conclusions because the possible glycosylation site is at the extracellular surface of FLVCR1, far from the ligand binding site. Moreover, our simulations focused on the release mechanism from the inward-facing structure, where glycosylation is less likely to influence the intracellular transport dynamics. In addition, our entry simulations were performed for the outward-facing structure of FLVCR2, which appears to lack a glycosylation site. It is important to note that no entry simulations were conducted on FLVCR1, the system in which glycosylation could potentially have a more pronounced effect.

The lipid bilayer composition used is physiologically relevant(?) It is well-known that lipid bilayer composition can affect the conformational dynamics of proteins (e.g. GPCRs).

- We agree with the reviewer's comment regarding the impact of lipid bilayer composition on protein dynamics and appreciate the opportunity to clarify the physiological relevance of the lipid composition used in our simulations. The specific composition was chosen to reflect the proteoliposome composition employed in transport assays, supported by several precedents in the literature where similar lipid compositions were used to study conformational dynamics in proteoliposomes. For example, similar membrane compositions have been reported in the context of functional studies of other human membrane proteins, as detailed in the following publications:

“Structure of hepcidin-bound ferroportin reveals iron homeostatic mechanisms”
(DOI: 10.1038/s41586-020-2668-z)

“Cryo-EM structure of PepT2 reveals structural basis for proton-coupled peptide and prodrug transport in mammals”
(DOI: 10.1126/sciadv.abh3355)

“Mechanism of Ca²⁺ transport by ferroportin”
(DOI: 10.7554/eLife.82947)

Reviewer Reports on the First Revision:

Referees' comments:

Referee #1 (Remarks to the Author):

I thank the authors for the thorough revision of their manuscript. All of my concerns have been properly addressed.

Referee #2 (Remarks to the Author):

The revised manuscript has been significantly improved. The new data support the proposed binding mode of ethanolamine by this transporter. I commend the authors for their efforts to improve the quality of the map and the model, which has strengthened the manuscript. I support this study for publication.

Referee #3 (Remarks to the Author):

Authors have satisfactorily answered to all my questions. A minor point - authors may want to double check the supplementary video numbering in the text.

Vivek Sharma

Author Rebuttals to First Revision:

Referees' comments:

Referee #1 (Remarks to the Author):

I thank the authors for the thorough revision of their manuscript. All of my concerns have been properly addressed.

- We greatly appreciate reviewer's acknowledgment of the efforts we made in revising the manuscript. The initial feedback from the reviewer was invaluable to us, and we are pleased to hear that we have successfully addressed all of the concerns.

Referee #2 (Remarks to the Author):

The revised manuscript has been significantly improved. The new data support the proposed binding mode of ethanolamine by this transporter. I commend the authors for their efforts to improve the quality of the map and the model, which has strengthened the manuscript. I support this study for publication.

- We are appreciative of the positive comments regarding the revised manuscript. It is encouraging to know that the additional data presented have solidified the proposed binding mode of ethanolamine by the transporter and that the efforts to enhance the map and model quality have strengthened the study.

The constructive feedback from the reviewer has been a significant factor in these improvements, and we are grateful for the support and recommendation for publication.

Referee #3 (Remarks to the Author):

Authors have satisfactorily answered to all my questions. A minor point - authors may want to double check the supplementary video numbering in the text.

- We thank the reviewer for the thoughtful review and for confirming that all of your questions have been satisfactorily addressed. We have corrected the numbering of the supplementary videos in the text and double-checked for accuracy.

The reviewer's insights have been invaluable to us throughout the revision process.